# Gustatory thalamic neurons mediate aversive behaviors

Feng Cao [1,2,4] ✉, Sekun Park [1,2], Jordan L. Pauli[1,2], Eden Y. Seo[1,2], An-Doan Nguyen [2] & Richard D. Palmiter [1,2,3] ✉

The parvicellular part of the ventral posteromedial nucleus (VPMpc) of the thalamus, also known as the gustatory thalamus, receives input from the parabrachial nucleus and relays taste sensation to the gustatory (or insular) cortex. Prior research has focused on the role of the VPMpc in relaying taste signals. Here we provide evidence showing that VPMpc also mediates aversive behaviors. By recording calcium transients in vivo from single neurons in mice, we show that neurons expressing cholecystokinin and the mu-opioid receptor in the VPMpc respond to various noxious stimuli and fear memory. Chemogenetic and optogenetic activation of these neurons enhances the response to aversive stimuli, whereas silencing them attenuates aversive behaviors. The VPMpc neurons directly innervate neurons in the insular cortex and rostral lateral amygdala. This study expands the role of the VPMpc to include transmitting aversive and threatening signals to the insular cortex and lateral amygdala.

The thalamus is considered as one of the most highly interconnected brain regions, acting as a hub for relaying sensory information between different subcortical areas and the cerebral cortex[1–3]. It is composed of different thalamic nuclei and segmented into several subregions based on anatomical location and thalamic-cortical connectivity patterns[3–5]. Each subregion serves an important role, ranging from relaying sensory and motor stimulations to regulation of sleep, consciousness, and learning and memory[2–5].

The parvicellular part of ventral posteromedial nucleus (VPMpc), also called the gustatory thalamus, is located medial to the ventral posteromedial nucleus (VPM) and contains neurons that process and transmit taste information to the insular cortex (IC)[6–8]. In the rodent gustatory sensory pathway, taste-related information from the tongue travels through cranial nerves VII, IX and X, converges in the nucleus solitarius (NTS) in the medulla, sends ascending neural projections to the parabrachial nucleus (PBN) in the dorsal pons, which projects to the VPMpc, and then to the gustatory part of the IC[6,9–13]. Neurons in VPMpc reliably encode the physiochemical identity of gustatory stimuli from different tastants delivered to the tongue and exert a strong influence on IC activity[7,14,15]. A few early studies found that VPMpc

neurons also respond to tactile and thermal stimuli on the tongue of anesthetized rats[8,16]. Neurons in VPMpc participate in taste expectations; lesions of the VPMpc in rats disrupted aversive and appetitive anticipatory taste[17], while an expected anticipatory cue improved taste processing by VPMpc neurons[18].

Although the VPMpc is widely associated with gustatory function, little is known about its involvement in other neurological functions or innate behaviors. The VPMpc receives inputs from several different neuron clusters in PBN including *Satb2*-expressing neurons (encoding special AT-rich sequence-binding protein 2) in the lateral and waist regions of the PBN and *Calca*-expressing neurons (encoding calcitonin gene-related peptide, CGRP) in the external lateral PBN[19,20]. The *Satb2* neurons have been shown to relay taste information to the VPMpc, but they project to many additional brain regions that are not known to be involved in taste processing[20,21]. The CGRP[PBN] neurons are activated by most sensory modalities (e.g., visceral malaise, taste, temperature, pain, itch) and function as a general alarm[22–24]. Photoactivation of the CGRP[PBN] axon terminals in VPMpc elicited freezing, fear memory and avoidance behavior, while photoinhibition of the VPMpc terminals alleviated fear response and

[1]Howard Hughes Medical Institute, University of Washington, Seattle, WA, USA. [2]Departments of Biochemistry, University of Washington, Seattle, WA, USA. [3]Department of Genome Science, University of Washington, Seattle, WA, USA. [4]Present address: Behavioral Neuroscience Program, Department of Psychology, Western Washington Universiity, Bellingham, WA, USA. ✉e-mail: feng.cao@wwu.edu; palmiter@uw.edu

fear memory[25]. Among all the brain regions that receive sensory input from CGRP[PBN] neurons, VPMpc exhibited significantly greater excitation from terminal activation than any other recorded projection sites[25]. Recent studies also found that the gustatory cortex encodes aversive taste memory and aversive states[26–28]. Given the interconnectivity of VPMpc with PBN and IC, we hypothesized that VPMpc also transmits aversive signals to the IC and mediates aversive behaviors.

Here we provide anatomical and functional evidence demonstrating that CGRP[PBN] neurons innervate molecularly defined neurons in VPMpc neurons that project axons to IC and rostral lateral amygdala (rLA). Using genetic and viral strategies to deliver Cre-dependent genes to VPMpc neurons, we found that VPMpc neurons responded to multiple aversive stimuli and cues that predicted them. Together with the gain-of-function and loss-of-function studies, we demonstrate that VPMpc neurons bidirectionally contribute to behavioral responses to aversive events, providing evidence that VPMpc neurons process non-gustatory signals.

## Results

### Cholecystokinin (*Cck*) and mu opioid receptor (*Oprm1*) genes are co-expressed in VPMpc neurons that are directly innervated by CGRP[PBN] neurons

Prior experiments revealed extensive axonal projections to the VPMpc by CGRP[PBN] neurons labeled with fluorescent probes and photoactivation of axon terminals in the VPMpc elicited significant biological effects[19,25]. To gain direct access to the VPMpc neurons that are targets of the CGRP[PBN] neurons, we sought to identify useful Cre-driver lines of mice. We initially checked Cre-driver lines of mice for receptors of the neuropeptides made by CGRP[PBN] neurons (CGRP, NTS, PACAP and TAC1) by injecting AAV carrying Cre-dependent fluorescent proteins into the VPMpc of *Calcrl^Cre, Ntsr1^Cre, Adcyap1r1^Cre* or *Tacr1^Cre* mice, respectively (Supplementary Fig. 1a–d); however, none of these Cre-driver lines expressed fluorescent protein within the VPMpc (Supplementary Fig. 1e–h). We then turned to Allen Mouse Brain Altas[29] to look for candidate genes expressed in the VPMpc for which Cre-driver lines of mice existed. We focussed on two gene candidates: *Cck* and *Oprm1*. To verify their expression in VPMpc, AAV1-DIO-YFP virus was injected into VPMpc of *Cck^Cre* or *Oprm1^Cre* mice (Fig. 1a); YFP was expressed in the VPMpc of both lines of mice (Fig. 1b).

To determine whether CCK[VPMpc] or OPRM1[VPMpc] neurons are downstream of CGRP[PBN] neurons, we injected AAV carrying Cre-dependent hM3Dq:mCherry, an excitatory DREADD (Designer Receptors Activated Only by Designer Drugs)[30,31] or just mCherry as control into the PBN of *Calca^Cre* mice (Fig. 1c). After allowing 6 weeks for viral expression, hM3Dq was activated by injection of its ligand, clozapine-N-oxide (CNO, 1 mg/kg, i.p.) and the brains were prepared 45 min later for in situ hybridization using RNAscope probes for *Cck, Oprm1* and *Fos*. *Cck* mRNA was colocalized with *Oprm1* mRNA in nearly all VPMpc neurons (Fig. 1d–f), which were about 50% of all DAPI-positive cells in VPMpc (Fig. 1g). After CNO administration, *Fos* was expressed in >18% of either the *Cck*- or *Oprm1*-expressing neurons in mice with hM3Dq:mCherry virus and given CNO, but in <0.6% of cells when CNO was injected in mice with the control mCherry virus (Fig. 1f, h). The percentage of *Fos* mRNA colocalized with either *Cck* or *Oprm1* mRNA in VPMpc was >98%. We conclude that *Cck* and *Oprm1* are co-expressed in the same neurons, and many of them express *Fos* mRNA after activation of CGRP[PBN] neurons with hM3Dq and CNO. To verify the proportion of Cck/Oprm1-expressing VPMpc neurons within the VPMpc, we examined the colocalization of *Cck* mRNA and *Slc17a6* (encodes Vglut2) mRNA in VPMpc neurons in wild-type mice. As shown in Supplementary Fig. 2a, b, *Cck* mRNA was colocalized with *Slc17a6* mRNA in nearly all VPMpc neurons, which are considered to be exclusively glutamatergic[32,33].

We used electrophysiology and optogenetic methods to determine whether CCK[VPMpc] and OPRM1[VPMpc] neurons are directly innervated by the ChR2-expressing presynaptic terminals from CGRP[PBN] neurons. To specifically target the CCK[VPMpc] and OPRM1[VPMpc] neurons, we generated mice expressing *Calca^FLPo::Cck^Cre* or *Calca^FLPo:: Oprm1^Cre* and then injected AAV carrying FLPo-dependent ChR2:YFP into PBN and Cre-dependent mCherry into VPMpc (Fig. 1i). After allowing 6 weeks for viral expression, we prepared brain slices and recorded excitatory post-synaptic currents (EPSCs) in response to photoactivation in CCK[VPMpc] or OPRM1[VPMpc] neurons (i.e., mCherry-positive soma with nearby YFP fibers from *Calca* neurons, Fig. 1j, k). Blue-light, optical stimulation of CGRP[PBN] terminals in the VPMpc evoked EPSCs in both CCK[VPMpc] (13/13) and OPRM1[VPMpc] (41/43) neurons of *Calca^FLPo::Cck^Cre* and *Calca^FLPo::Oprm1^Cre* mice, with a <3-ms onset latency (Fig. 1l). There was no significant difference between the evoked EPSC amplitude in CCK[VPMpc] and OPRM1[VPMpc] neurons (Fig. 1m). Bath application of μ-amino-3-hydroxy-5-methyl-4-isoxazolepropionic (AMPA) receptor antagonist NBQX (10 μM) and N-methyl-D-aspartate (NMDA) receptor antagonist (2R)-amino-5-phosphonovaleric acid (APV, 50 μM) to the brain slices completely blocked the EPSCs (Supplementary Fig. 3a, b). We also investigated blue-light induced synaptic release from isolated ChR2-transduced PBN axons terminals in the VPMpc. Bath application of the Na$^+$ channel blocker, tetrodotoxin (TTX, 1 μM), abolished light-evoked EPSCs which could be reinstated by the subsequent addition of 1 mM K$^+$ channel blocker 4-aminopyridine (4AP) (Supplementary Fig. 3c–e), although the onset latency was typically delayed[34,35]. These results indicate that the light-evoked EPSCs in CCK[VPMpc] or OPRM1[VPMpc] are mono-synaptic, glutamatergic and action-potential dependent. Thus, we conclude that *Cck* and *Oprm1* genes are co-expressed in nearly all VPMpc neurons, and these neurons receive direct, excitatory input from CGRP[PBN] neurons.

### VPMpc neurons respond to aversive sensory modalities

CGRP[PBN] neurons serve as a general alarm and are activated by many aversive events and cues that predict them[22,24]. Because CGRP[PBN] neurons innervate CCK and OPRM1 neurons in VPMpc, we hypothesized that they would also respond to aversive stimuli. We used *Cck^Cre* mice and 1-photon calcium imaging to evaluate whether VPMpc neurons respond to aversive stimuli. *Cck^Cre* mice were injected with AAV carrying a Cre-dependent calcium indicator, GCaMP6m; then a gradient refractive index (GRIN) lens was implanted over the VPMpc to monitor the CCK[VPMpc] neuron activity in freely moving mice (Fig. 2a). We examined the responses to a 90-dB hand clapping sound (Fig. 2b), a 1-s air puff onto the head (Fig. 2e), a 1-s tail pinch (Fig. 2h), or 10 s of lifting the mice out of the cage by their tail (Fig. 2k). We applied these different sensory modalities to mice individually and recorded the calcium signal before and after each stimulus (78 neurons from 4 mice). The 90-dB hand clapping sound generated a smaller increase in calcium activity (Fig. 2b–d) than either the air puff (Fig. 2e–g) or tail pinch (Fig. 2h–j). Lifting the mouse by the tail also elevated the calcium activity with a gradual increase during the 10-s lifting period, which remained elevated for 10 s after the mice were returned to the cage (Fig. 2k–m).

We applied three different shock intensities (0.15, 0.3 and 0.5 mA) to mice placed on a shock grid and recorded the calcium activity generated by each intensity. All shock intensities generated increases in calcium activity that were positively correlated with shock intensities (Fig. 2n–s). The 0.5-mA shock activated the largest number of neurons (Fig. 2r, s), and most of them were also activated by the 0.3-mA shock (Fig. 2q, r). We conclude that VPMpc neurons respond to many aversive modalities.

Because VPMpc neurons are well known to relay gustatory signals[6–8,13], we recorded CCK neuron activity using the same viral strategy (55 neurons from 2 mice) with 5-s exposure to water, 5% sucrose or 0.1 mM quinine. Licking for each of these tastes increased

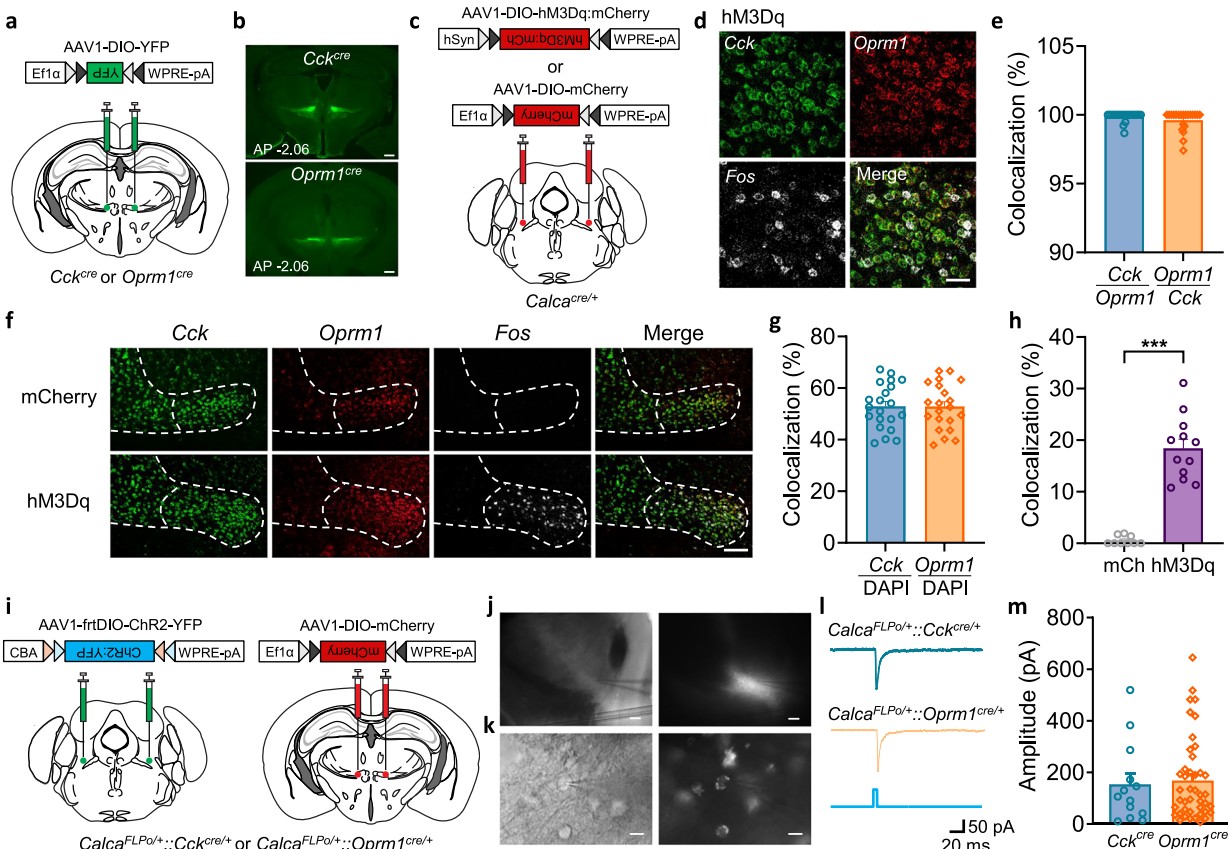

**Fig. 1 | CCK^VPMpc neurons and OPRM1^VPMpc neurons are directly innervated by CGRP^PBN neurons. a** Scheme showing bilateral injections of AAV1-DIO-YFP into the VPMpc of *Cck^Cre* or *Oprm1^Cre* mice. **b** AAV1-DIO-YFP expression in VPMpc of *Cck^Cre* or *Oprm1^Cre* mice, scale bar 500 μm. **c** Scheme showing bilateral injections of AAV1-DIO-mCherry or AAV1-DIO-hM3Dq:mCherry into the PBN of *Calca^Cre* mice. **d** High magnification images showing the overlap of *Cck*, *Oprm1*, and *Fos* mRNA in VPMpc of mice injected AAV1-DIO-hM3Dq:mCherry into PBN, scale bar 40 μm. **e** Summary data of the *Cck* and *Oprm1* colocalization (*n* = 21 per group, two-tailed Mann–Whitney test, *P* = 0.1917). **f** Low magnification images showing *Cck*, *Oprm1*, and *Fos* mRNA in VPMpc of mice injected AAV1-DIO-mCherry and AAV1-DIO-hM3Dq:mCherry into PBN, scale bar 150 μm. **g** Summary data showing the percentage of DAPI- positive cells that express *Oprm1* or *Cck* mRNA (*n* = 21 per group, unpaired two-tailed *t* test, *P* = 0.9677). **h** Summary data of *Fos* and *Cck* mRNA colocalization in both mCherry (mCh, *n* = 9) and hM3Dq (*n* = 12) groups (Two-tailed Mann–Whitney test, *P* < 0.0001). **i** Scheme showing bilateral injections of AAV1-frtDIO-ChR2:YFP or AAV1-DIO-mCherry into the PBN and VPMpc of *Calca^FLPo::Cck^Cre* and *Calca^FLPo::Oprm1^Cre* mice. **j** Image of VPMpc under 4× differential interference contrast (DIC) objective (left) and fluorescence object (right) with recording electrode on top of it, scale bar 150 μm. **k** Image of VPMpc neurons under 40× DIC objective (left) and fluorescence object (right) with recording electrode attached to neuron, scale bar 10 μm. **l** Sample traces and **m** summary figure of 470-nm, blue-light pulse of light-evoked EPSCs in the VPMpc neurons of *Calca^FLPo::Cck^Cre* (*n* = 13) and *Calca^FLPo::Oprm1^Cre* (*n* = 43) mice (Two-tailed Mann–Whitney test, *P* = 0.7013). Circled data points represent *Cck* or *Cck^Cre* mice, diamonds represent *Oprm1* or *Oprm1^Cre* mice in the summarized bar graphs. Data are represented as mean ± SEM. ****P* < 0.001.

calcium activity in water-restricted mice (Supplementary Fig. 4a–f). Because we observed increased activity to both aversive and taste stimuli, we asked whether the same or different VPMpc neurons respond to them. By alternately applying three taste stimuli (water, sucrose and quinine) and four aversive sensory stimuli (air puff, clapping, tail lifting and tail pinch) to mice, we recorded responses of individual neurons to these seven stimulus modalities (Supplementary Fig. 5a). We also quantified the absolute value of each response, normalized by the maximum response of each stimulus (Supplementary Fig. 5b). Out of 39 recorded neurons, 38 neurons responded to at least one stimulus. Many (31) neurons responded to more than one stimulus modality. Some neurons that responded to taste stimuli also responded to aversive sensory modalities. This analysis suggests that individual VPMpc neurons are not tuned to respond to a specific stimulus modality (Supplementary Fig. 5b, c).

## Fear memory activates VPMpc neurons

We analyzed CCK^VPMpc neuronal activity during auditory fear conditioning, an associative learning paradigm in which a tone, a conditioned stimulus (CS), was paired with a foot shock, an unconditioned stimulus (US). We used the same mice with GCaMP6m expression and GRIN lens implantation as described above and measured the calcium dynamics of CCK^VPMpc neurons while performing a three-day, fear conditioning protocol (Fig. 3a–c). On Day 1, only a few CCK^VPMpc neurons were activated when mice were exposed to 10 trials of a 10-s tone (Fig. 3d, g, Supplementary Fig. 6). On Day 2, with 10 trials of the CS-US association, many CCK^VPMpc neurons were activated by the foot shock (Fig. 3e, h, Supplementary Fig. 7), which is consistent with the calcium activity change under foot shock in the previous experiment (Fig. 2o). On Day 3, the mice were exposed to the tone alone in a novel context; many CCK^VPMpc neurons responded during the 10-s tone, and the activity persisted beyond the 10 s of recording following the tone (Fig. 3f, i, Supplementary Fig. 8); the response was much greater than the response on Day 1. Compared to the activity change during Day 1 (Fig. 3j), the proportion cells responding on both Day 2 (Fig. 3k) and Day 3 (Fig. 3l) was significantly increased. These calcium-imaging studies reveal that CCK^VPMpc neurons become activated by the CS after conditioning and reflect the recall of the fear memory, like that observed in CGRP^PBN neurons[22].

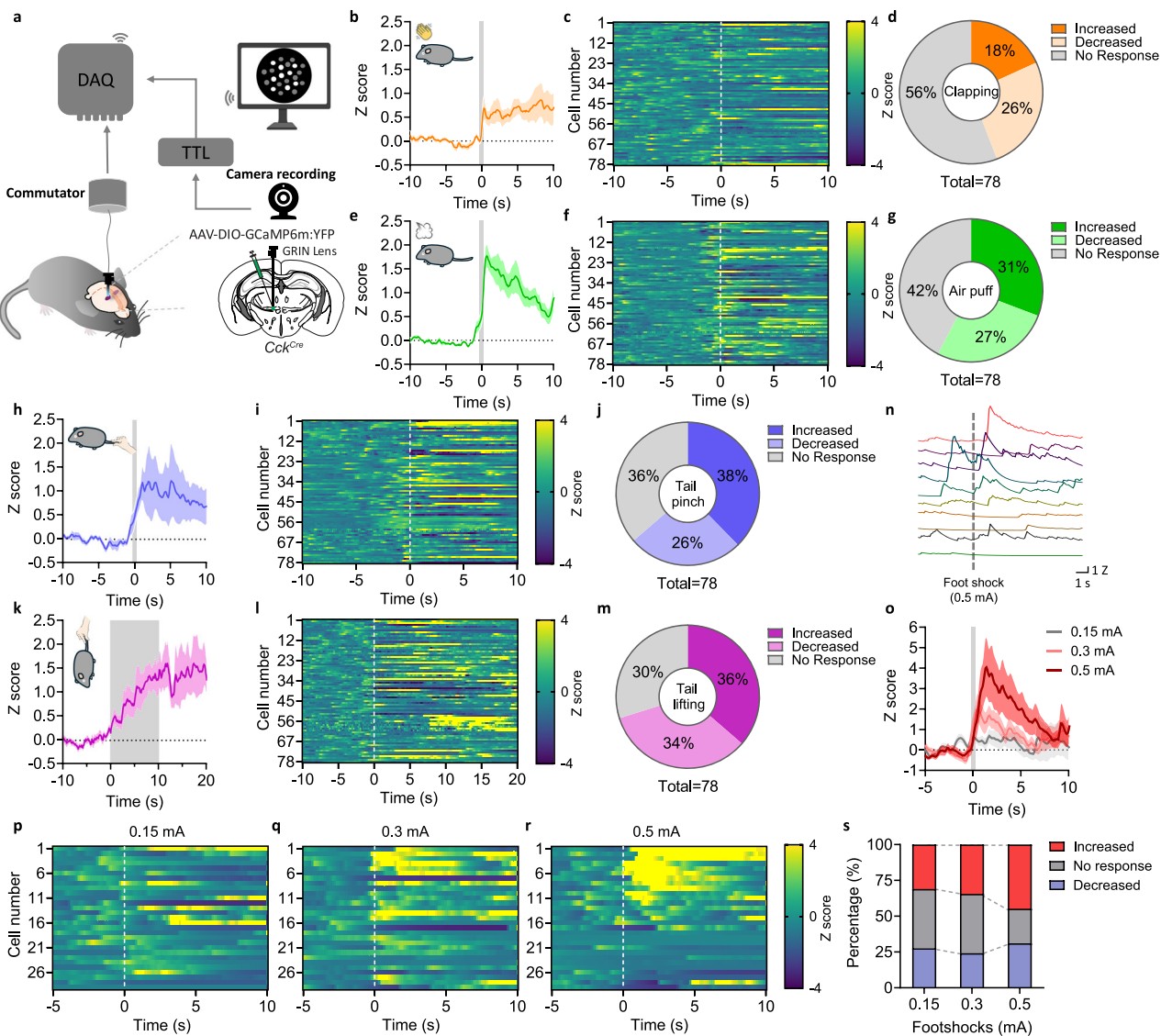

**Fig. 2 | VPMpc neurons respond to aversive sensory stimuli. a** Scheme of 1-photon, single-cell calcium imaging in freely moving mice. **b–m** Diagram and average traces of all recorded neurons (**b**, **e**, **h**, **k**), heat maps of individual neuronal responses (**c**, **f**, **i**, **l**), percentage of increased, decreased, and no response CCK^VPMpc neurons (**d**, **g**, **j**, **m**) in response to hand clapping sound, air puff, tail pinch, and tail lifting. Vertical dashed lines: onset of each sensory stimulus. **n** Sample calcium fluorescence activity trace of 10 CCK^VPMpc neurons when mice underwent 0.5-mA foot shock. **o** Average traces of all recorded neurons, **p–r** heat maps of individual CCK^VPMpc neuron calcium fluorescence activity in response to 0.15-, 0.3- and 0.5-mA foot shocks. Vertical dashed lines: onset of the foot shocks. **s** Percentage of increased, decreased, and no response CCK^VPMpc neurons in response to 0.15-, 0.3-, and 0.5-mA foot shocks. Data are represented as mean ± SEM.

## Inactivation of CCK/OPRM1^VPMpc neurons attenuates the response to aversive stimuli

To determine whether CCK/OPRM1^VPMpc neurons are necessary for the aversive behavioral responses, we silenced them with tetanus toxin (TetTox), which cleaves synaptic vesicle protein synaptobrevin, thereby preventing neurotransmitter release and signaling to post-synaptic neurons[36,37]. *Oprm1^Cre* or *Cck^Cre* mice were bilaterally injected in the VPMpc with AAV carrying Cre-dependent tetanus toxin (AAV$_{DJ}$-EF1a-DIO-GFP:TetTox) or YFP (AAV1-DIO-YFP) as control (Fig. 4a). Note that many subsequent experiments were performed with cohorts of *Cck^Cre* (circles) or *Oprm1^Cre* (diamonds) mice and the results were combined because these genes are co-expressed in VPMpc neurons. We first examined tactile sensitivity to a set of von Frey filaments with different stiffness. As shown in Fig. 4b, mice injected with the TetTox virus manifested a significant increase in the paw-withdraw threshold response measured by von Frey filaments, indicating that silencing

them decreases tactile sensitivity. We also observed that mice with TetTox had fewer nocifensive behaviors and diminished escape attempts during exposure to noxious heat compared to control mice with only YFP (Fig. 4c, d).

We also tested whether inhibiting these neurons affected fear memory formation in the 3-day fear conditioning paradigm described above (Fig. 4e). Mice injected with the control virus had an increased freezing response to the shock during the training trials, whereas the response of the TetTox-expressing mice was much weaker (Fig. 4f); the response to the tone alone in a novel box the next day was also weaker (Fig. 4f). The control (YFP) mice had a strong freezing response to the original box when tested the next day, but the TetTox-expressing mice did not (Fig. 4g). We also examined the response of the TetTox mice in a 3-day, fear-potentiated startle test, in which a light predicted a foot shock on Day 2, and we measured the startle response to the light on Day 3 (Fig. 4h). The startle responses to a range of pulse intensities

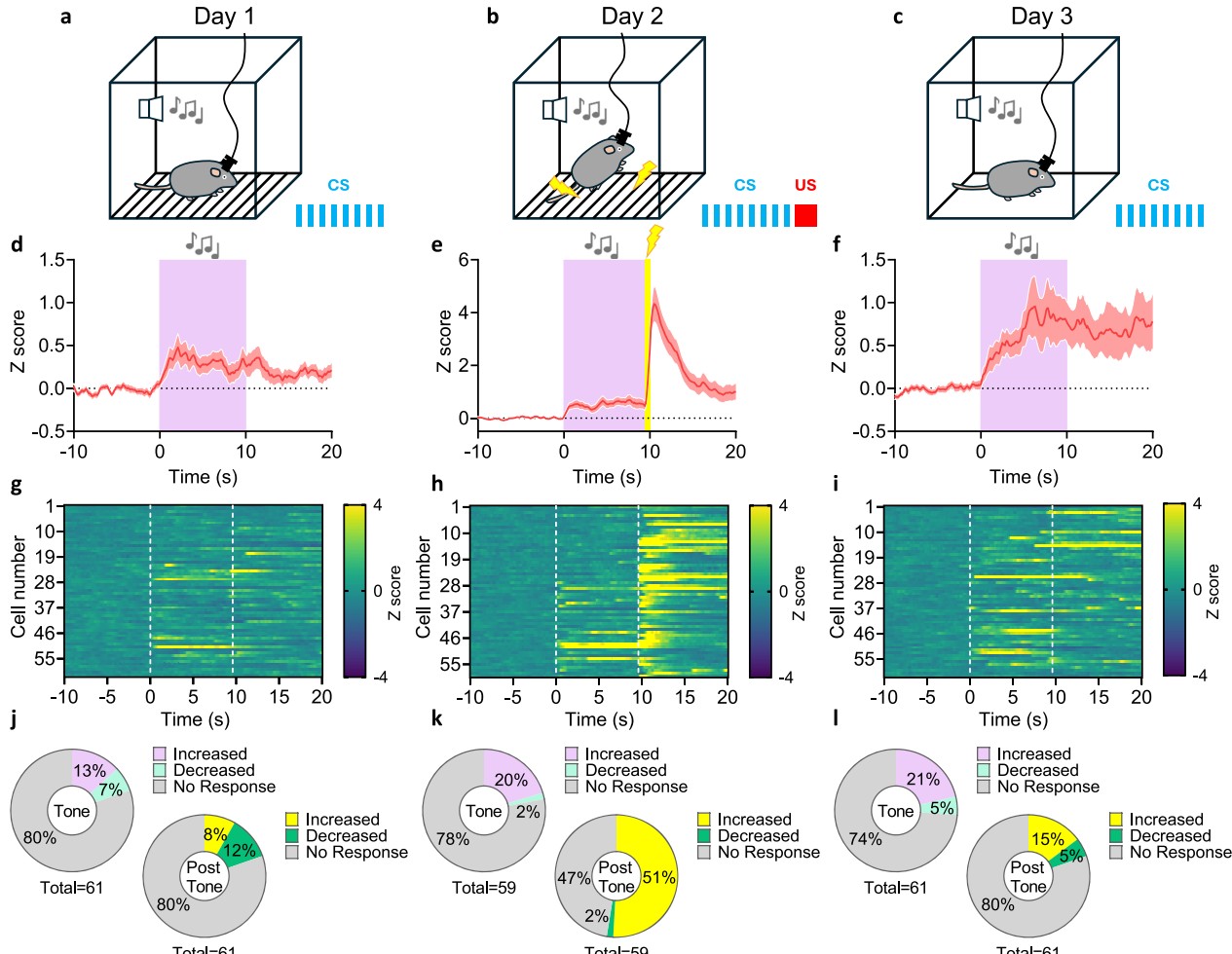

**Fig. 3 | VPMpc neurons contribute to fear-learning memory recall.**
**a–c** Schematic diagram of calcium imaging in 3-day fear conditioning paradigm. Average traces of all recorded neurons (**d**–**f**) and heat maps of individual neuronal responses (**g**–**i**) of CCK^VPMpc neuron calcium fluorescence activity 10 s before tone, 10 s during tone, 10 s after tone on Day 1, Day 2 and Day 3. Tone starts at 0 s in all traces, shock starts at 9.5 s on Day 2. Vertical dashed lines: onset of each tone and foot shock. Percentage of increased, decreased and no response CCK^VPMpc neurons during 10-s tone (left) and 10-s post-tone (right) on Day 1 (**j**), Day 2 (**k**) and Day 3 (**l**). Data are represented as mean ± SEM.

were similar between the two groups of mice before training (Fig. 4i); however, after training, the light (CS cue) enhanced the startle response in the YFP (control) group but not in the TetTox group (Fig. 4j). These results confirm that the TetTox- expressing mice have difficulty learning or remembering the CS-US association.

To ascertain the affective state generated by silencing CCK/OPRM1^VPMpc neurons, we used a conditioned place aversion (CPA) test with two chambers that had either stripes or spots on the wall (Fig. 4k). On the pre-test day, both YFP and TetTox-expressing mice exhibited equal preference towards the two chambers (Fig. 4l, m). After pairing one chamber with U50488 (U50, 5 mg/kg), a kappa opioid receptor agonist that is aversive[38,39], mice injected with YFP demonstrated an aversion to U50, whereas mice injected with TetTox did not (Fig. 4l, m). Taken together, these results reveal that inactivation of CCK/OPRM1^VPMpc neurons interfered with the normal responses to aversive stimuli.

## Activation of CCK/OPRM1^VPMpc neurons enhances the response to aversive stimuli
We next examined whether activation of CCK/OPRM1^VPMpc neurons could alter the response to aversive stimuli. We bilaterally injected AAV carrying Cre-dependent hM3Dq:mCherry (or mCherry as control) into VPMpc of *Oprm1^Cre* or *Cck^Cre* mice (Fig. 5a). The effect of hM3Dq

activation by CNO was verified in brain slices using cell-attached, voltage clamp to measure spontaneous cell firing before and after CNO application. Bath application of CNO increased spontaneous cell firing in neurons with hM3Dq:mCherry expression, but not neurons with only mCherry expression (Fig. 5b, c). We examined whether activation of CCK/OPRM1^VPMpc neurons changed tactile sensitivity to von Frey filaments of different stiffness. The hM3Dq-expressing mice treated with CNO had a significant decrease in the paw-withdraw threshold (Fig. 5d), indicating that activating these neurons induced allodynia. We also found that mice with hM3Dq:mCherry had more nocifensive responses and increased escape attempts during exposure to noxious heat compared with mice with mCherry (Fig. 5e, f). These results indicate that chemogenetic activation of CCK/OPRM1^VPMpc neurons enhances the nocifensive response to noxious stimuli.

To further examine the effect of activating CCK/OPRM1^VPMpc neurons, we unilaterally injected AAV carrying Cre-dependent channelrhodopsin, ChR2:YFP (or YFP as control) into VPMpc of *Cck^Cre* and *Oprm1^Cre* mice and fiber-optic cannulas were placed over the VPMpc (Fig. 5g). We placed both groups of mice in an open-field chamber and recorded their mobility throughout eleven 30-s time bins with 473-nm blue light stimulation (10 Hz, 10 ms, 2 mW) during time bins 3, 6 and 9 (Fig. 5h). Mice expressing ChR2 exhibited higher travel distance in each light-stimulated bin, whereas control mice had the same travel

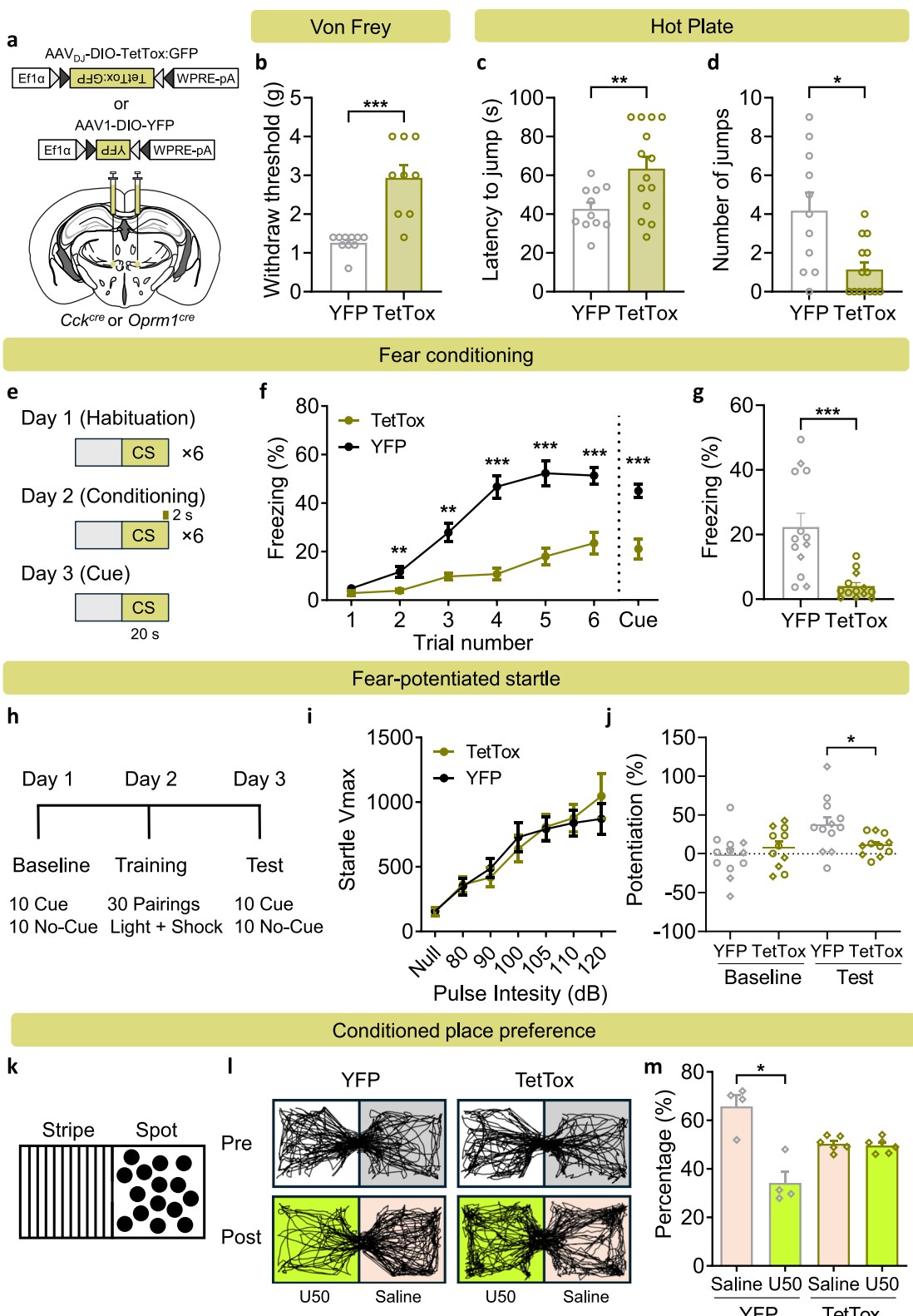

distance across all bins (Fig. 5h, i). We also examined the tactile sensitivity in both control and ChR2 groups after they received 5-min of photo-stimulation (10 Hz, 10 ms, 2 mW). Photoactivating ChR2-expressing CCK/OPRM1$^{VPMpc}$ neurons had the same allodynia effect as activating hM3Dq-expressing VPMpc neurons with CNO (Fig. 5d, j). Interestingly, the allodynia caused by photo-stimulation lasted 2 days (Fig. 5j), indicating that brief activation of CCK/OPRM1$^{VPMpc}$ neurons

can trigger prolonged tactile sensitivity. Additionally, we used 2-s photo-stimulation (30 Hz, 10 ms, 15 mW) to replace the traditional foot shock in the fear conditioning paradigm but found that directly activating the CCK/OPRM1$^{VPMpc}$ neurons was unable to serve as an unconditioned stimulus (Supplementary Fig. 9a–c).

To determine the affective state when activating CCK/OPRM1$^{VPMpc}$ neurons, we used a real-time, place-preference (RTPP) test to ask

**Fig. 4 | Inactivation of CCK/OPRM1^VPMpc neurons attenuates aversive responses.** **a** Bilateral injections of AAV_DJ-DIO-TetTox-GFP and AAV1-DIO-GFP into the VPMpc of *Cck^Cre* or *Oprm1^Cre* mice. **b** Von Frey test showing increased withdraw threshold in TetTox-injected mice (YFP: $n = 10$, TetTox: $n = 9$, two-tailed Mann–Whitney test, $P < 0.0001$). Hot plate test showing increased latency to first jump (Unpaired two-tailed t test, $P = 0.0071$) (**c**) and decreased number of jumps (Two-tailed Mann–Whitney test, $P = 0.0254$) (**d**) in TetTox-expressing mice (YFP: $n = 11$, TetTox: $n = 14$). **e** Experimental paradigm for 3-day, cue-dependent, fear-conditioning test. Summary data showing decreased freezing responses of TetTox-expressing mice during training (Two-way repeated measures ANOVA with Holm-Sidak's multiple comparisons, $F_{(1, 24)} = 66.93$, $P < 0.0001$), the cue test (Unpaired two-tailed t test, $P = 0.0001$) (**f**) and in context test ($n = 13$ per group, two-tailed Mann–Whitney test, $P < 0.0001$) (**g**). **h** Experimental paradigm for 3-day fear-potentiated startle test. **i** Startle test showing no difference between startle responses to multiple pulse intensities in YFP- and TetTox-expressing mice (YFP: $n = 12$, TetTox: $n = 14$; two-way repeated measures ANOVA with Holm-Sidak's multiple comparisons, $F_{(1, 24)} = 0.0122$, $P = 0.9128$). **j** Higher conditioned startle response after light cue training in YFP mice ($n = 12$) but not in TetTox mice ($n = 11$) (Baseline: unpaired two-tailed t test, $P = 0.3914$; Test: unpaired two-tailed t test, $P = 0.0314$). **k** Scheme showing conditioned place preference chamber with stripes on one side and spots on the other. **l** Representative mobility traces showing no preference before U50-paired conditioning (Day 1), higher preference in Saline chamber after conditioning (day 4) in YFP but not TetTox mice. **m** Summary data showing aversion to U50 in YFP mice ($n = 4$) but not TetTox mice ($n = 6$) after conditioning (YFP: paired two-tailed t test, $P = 0.0426$; TetTox: paired two-tailed t test, $P = 0.7729$). Circled data points represent *Cck^Cre* mice, diamonds represent *Oprm1^Cre* mice in the summarized bar graphs. Data are represented as mean ± SEM. *$P < 0.05$, **$P < 0.01$, ***$P < 0.001$.

whether mice prefer or avoid optical stimulation. A two-chambered box with one side blank and the other decorated with spots was used for these experiments (Fig. 5k). The mice could freely explore the two chambers for 10 min on Day 1. Both groups of mice demonstrated similar preferences towards the two chambers without optical stimulation on Day 1 (Fig. 5l, m). We then paired optical stimulation (10 Hz, 10 ms, 2 mW) in only one of the two chambers. Mice with only YFP expression had no preference for either side, whereas mice with ChR2 stimulation robustly avoided the side with optical stimulation (Fig. 5l, n), indicating that photoactivation of CCK/OPRM1^VPMpc neurons is aversive. We chose these relatively mild stimulation conditions because more robust stimulation (30 Hz, 10 ms, 10 mW) resulted in rotation behavior and body distortion (Supplementary Movies 1 & 2). Taken together, these results demonstrate that both chemogenetic and optogenetic activation of CCK/OPRM1^VPMpc neurons enhances the responses to aversive stimuli.

## Axonal projections of CCK^VPMpc neurons
VPMpc neurons have long been known to transmit taste signals to the forebrain of different mammals[6,8,40,41]. The projection from the VPMpc to the IC is a major component of the gustatory pathway from the taste buds in the tongue to the gustatory cortex[2,7,41]. To identify the axonal projections from VPMpc, we injected AAV expressing Cre-dependent fluorescent proteins (AAV1-DIO-YFP and AAV1-DIO-synaptophysin:mCherry) into the VPMpc of *Cck^cre* mice. A small volume of virus (15 nl) was injected and only those mice with expression restricted to the VPMpc were used to examine axonal projections (Fig. 6a–c). To obtain the whole brain projection pattern, we cut the entire brain into 35-μm sections and imaged every third coronal brain section. The imaged sections were stitched together and registered to view the expression of YFP and mCherry throughout the brain (Supplementary Movie 3). Other than the strong projection to IC, we also observed a weaker projection to the lateral amygdala (LA) (Fig. 6d). After quantifying the signal intensity throughout the IC, we observed expression of YFP and mCherry in three collected sections (Bregma +0.75 to −0.4 mm), which covers the medial and posterior of IC (Supplementary Fig. 10a–c). We also observed two separate bands in different cortical layers; one of them was in layer 1 and the other larger band was in layers 3-4 (Supplementary Fig. 10b, c). The VPMpc projection to the LA (Bregma −1.3 to −1.8 mm) was restricted to the rostral portion of the LA (Supplementary Fig. 11a–c).

We used electrophysiology and optogenetic methods to determine whether VPMpc neurons make functional connections with IC and rLA. We injected AAV carrying Cre-dependent ChR2 (AAV1-Ef1a-DIO-ChR2:YFP) into the VPMpc of *Cck^cre* mice (Fig. 6e). After 6 weeks for viral expression, we prepared brain slices containing IC and rLA and recorded the activity of neurons adjacent to YFP-labeled axon terminals in these two brain regions (Fig. 6e, f). Light-evoked EPSCs were captured in both IC (23/23) and rostral LA (17/19) neurons (Fig. 6g). Although the amplitude of EPSCs in rLA were slightly smaller than those in the IC, there was no significant difference between them (Fig. 6g, h). Bath application of TTX (1 μM) blocked the EPSCs in both brain regions, which were restored by adding 4AP (1 mM) (Fig. 6i–n), suggesting both IC and LA receive direct monosynaptic inputs from CCK^VPMpc neurons. Thus, we conclude that CCK^VPMpc neurons send direct efferent projections to IC and rLA.

## Stimulation of VPMpc projections
There are two possible models of the CCK^VPMpc neuron connectivity to IC and rLA. Model A: parallel pathways originating from distinct cells in VPMpc to either IC or rLA. Model B: collateral pathways originating from a single VPMpc neuron that project to both IC and rLA (Fig. 7a). To distinguish between them, we employed retrograde tracing and INTRSECT virus strategy to selectively isolate subsets of CCK^VPMpc neuron and characterize their projections to target brain structures. We unilaterally injected AAV carrying retrogradely transported Flp-recombinase (AAVretro-Flp:zsGreen) into IC and the INTRSECT virus (AAV1-Ef1a-Cre_on Flp_on-mCherry) into VPMpc in the same hemisphere of *Cck^cre* mice (Fig. 7b). With the expression of Flp:zsGreen in the IC and the fluorescence tracer mCherry requiring both Cre and Flp for expression in VPMpc, we observed that the IC-projecting VPMpc neurons also send projections to LA (Fig. 7c), supporting model B with CCK^VPMpc neurons sending collaterals to both IC and rLA. Note that in this experiment a projection from the VPMpc to the neighboring amygdalostriatal transition area (ASt) and central nucleus of the amygdala (CeA) was also apparent (Fig. 7c).

To determine whether distinct output from the VPMpc mediate different aversive responses, we bilaterally injected VPMpc with AAV1-Ef1a-DIO-ChR2:YFP (or YFP as control) and implanted fiber-optic cannulas above the IC or rLA (Fig. 7d, k). Both ChR2 and YFP were expressed bilaterally in VPMpc, ranging from bregma −2.06 to −2.18 mm (Supplementary Fig. 12a, b). To measure the tactile sensitivity in both control and ChR2 groups, we used the same 5-min photoactivation protocol (10 Hz, 10 ms) but bilaterally stimulated the VPMpc terminals in IC or rLA with a higher light intensity (4 mW). One-time photoactivation of ChR2-expressing CCK^VPMpc neuron terminals in IC elicited allodynia that lasted 4 days (Fig. 7e). After recovery, we repeated the same stimulation for 5 consecutive days, which produced allodynia lasting 30 days (Fig. 7f). These results indicate that both the brief and repeated stimulation can trigger prolonged tactile sensitivity. We also used RTPP test to measure whether mice prefer or avoid optical stimulation of VPMpc-IC pathway. On Day 1, mice injected with YFP or ChR2:YFP had similar preference for both chambers without optical stimulation (Fig. 7g, h). On Day 2, mice with bilateral ChR2 stimulation of VPMpc-IC terminals spent less time in the side with optical stimulation while the control mice had no preference for either side (Fig. 7g, i). Additionally, ChR2-expressing mice continued to avoid the optical stimulation side after the light stimulation ceased (Fig. 7g, j), indicating that photoactivation of VPMpc-IC pathway is not only aversive but also generates a short-term aversive memory. We next

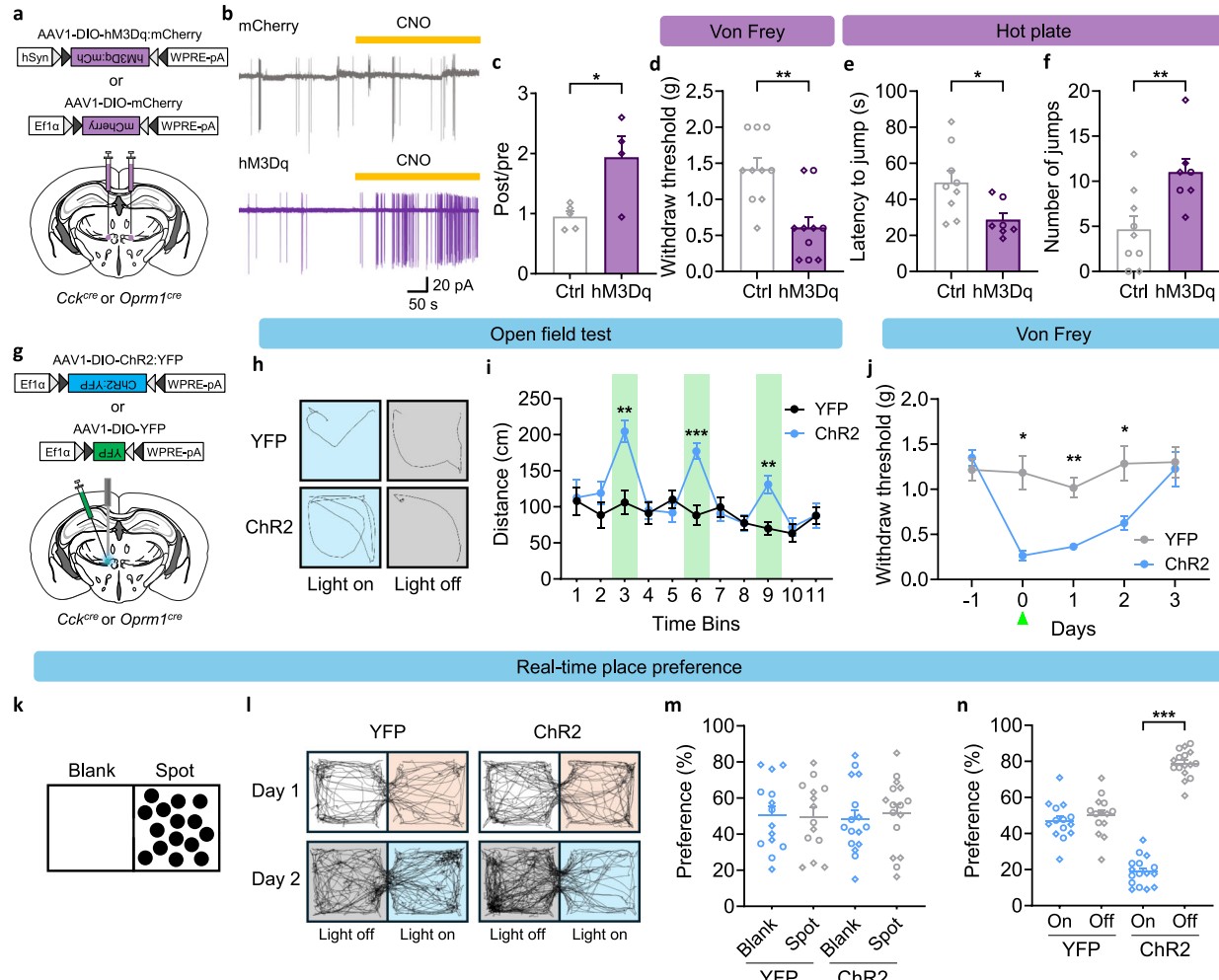

**Fig. 5 | Activation of CCK/OPRM1$^{VPMpc}$ neurons enhances aversive responses.**
**a** Bilateral injections of AAV1-DIO-hM3Dq:mCherry and AAV1-DIO-mCherry into the VPMpc. **b** Sample traces and **c** summary figure showing CNO-induced spontaneous cell firing in hM3Dq ($n = 4$) but not mCherry-injected ($n = 5$) mice (Unpaired two-tailed t test, $P = 0.0196$). **d** Von Frey test showing decreased paw withdrawal threshold in hM3Dq-injected mice ($n = 10$ per group, two-tailed Mann–Whitney test, $P = 0.0025$). Hot plate test showing decreased latency to jump (Unpaired two-tailed t test, $P = 0.0151$) (**e**) and increased number of jumps (Unpaired two-tailed t test, $P = 0.0098$) (**f**) in hM3Dq-injected mice (Ctrl: $n = 9$, hM3Dq: $n = 7$). **g** Unilateral injections of AAV1-DIO-ChR2:YFP, AAV1-DIO-YFP and optic fiber implantation above the VPMpc. **h** Representative mobility traces of mice with YFP and ChR2 during light on (bin 3) and light off (bin 4). **i** Open-field test showing increased distance traveled in ChR2 photo-stimulated (bin 3, 6, 9) mice ($n = 10$ per group, two-way repeated measures ANOVA with Holm-Sidak's multiple comparisons, F (1,

18) = 11.45, $P = 0.0033$). **j** Von Frey test showing continuous days of decreased paw-withdrawal threshold after 5 min of photo-stimulation in ChR2-expressing mice (YFP: $n = 6$, ChR2: $n = 8$, two-way repeated measures ANOVA with Holm-Sidak's multiple comparisons, F (1, 12) = 13.5, $P = 0.0032$). **k** Scheme of RTPP paradigm.
**l** Representative mobility traces showing no side preference in YFP and ChR2 mice without photostimulation on Day 1, higher preference in light-off chamber in ChR2-expressing but not in YFP mice on Day 2. **m** YFP mice ($n = 14$) and ChR2 mice ($n = 16$) had similar side preference without optical stimulation (YFP: paired two-tailed t test, $P = 0.9359$; ChR2: paired two-tailed t test, $P = 0.7615$). **n** Photo activation of ChR2- injected mice led to avoidance of light-paired side (YFP: paired two-tailed t test, $P = 0.6047$; ChR2: paired two-tailed t test, $P < 0.0001$). Circles represent $Cck^{Cre}$ mice, diamonds represent $Oprm1^{Cre}$ mice in the summarized bar graphs. Data are represented as mean ± SEM. *$P < 0.05$, **$P < 0.01$, ***$P < 0.001$.

measured the involvement of VPMpc-LA pathway in tactile sensitivity and the affective state. Like the sensitivity in VPMpc-IC pathway, one-time (5 min, 10 Hz, 10 ms, 4 mW) bilateral optical stimulation generated allodynia that persisted for 5 days (Fig. 7l) whereas 5 consecutive daily stimulations elicited allodynia lasting for 30 days (Fig. 7m). However, photo-activation of ChR2-expressing VPMpc-rLA terminals did not result in a preference for either side (Fig. 7n–q), indicating that only activating the projection to the IC is aversive in this assay.

## Discussion

We demonstrated that CCK$^{VPMpc}$ neurons and OPRM1$^{VPMpc}$ neurons are innervated by CGRP$^{PBN}$ neurons. Electrophysiological recordings provided functional evidence that the input from CGRP$^{PBN}$ neurons to CCK$^{VPMpc}$ or OPRM1$^{VPMpc}$ neurons is mono-synaptic and glutamatergic.

Calcium imaging in freely moving mice revealed that $Cck$ neurons in the VPMpc respond to different aversive stimuli including hand clap, air puff, tail pinch, tail lifting and foot shock. They also responded to a tone (conditional stimulus) after it was associated with a foot shock. Inactivation of CCK$^{VPMpc}$ or OPRM1$^{VPMpc}$ neurons produced deficits in their responses to aversive stimuli while their activation enhanced responses to aversive stimuli. We also revealed direct, collateral axonal projections from VPMpc neurons to neurons in the IC and rLA. Photo-activating either the VPMpc-IC or VPMpc-rLA projections enhanced tactile sensitivity but only stimulating VPMpc-IC pathway was aversive. Prior to our study, the VPMpc projection to IC focussed on gustatory functions.

Quantitative in situ hybridization data revealed that $Cck$ and $Oprm1$ are co-expressed in 98% of the neurons and they represent ~50%

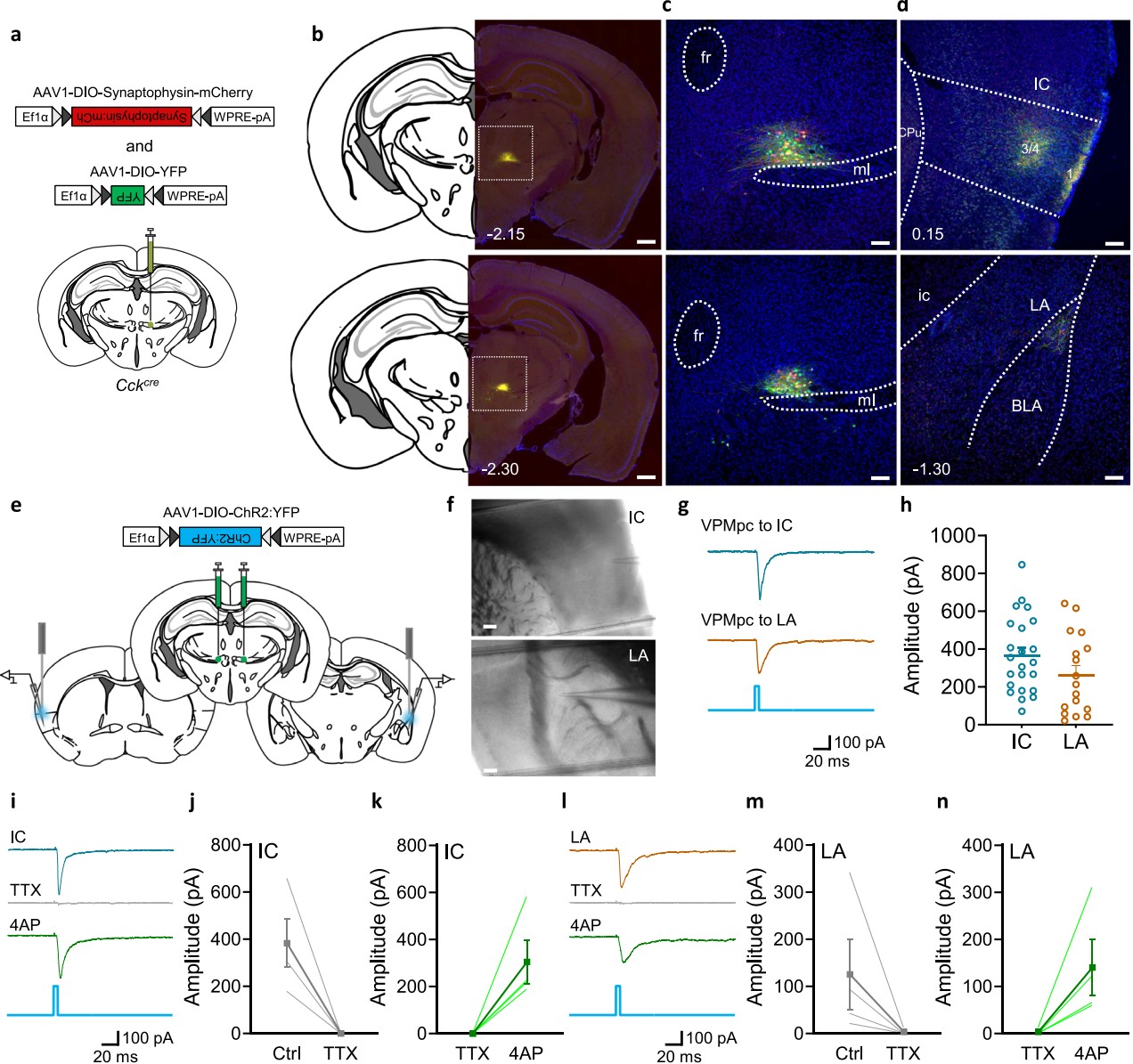

**Fig. 6 | CCK^VPMpc neurons send direct input to IC and LA. a** Scheme showing unilateral injection of AAV1-DIO-mYFP and AAV1-DIO-Synaptophysin:mCherry into the VPMpc of *Cck^Cre* mice. **b** YFP and Synaptophysin:mCherry expression in VPMpc of *Cck^Cre* mice, scale bar 500 μm. **c** Sample images showing higher magnification of YFP and Synaptophysin:mCherry expression in VPMpc of *Cck^Cre* mice, scale bar 100 μm. **d** Sample images showing VPMpc axon terminals in IC and LA, scale bar 100 μm. See Supplementary Figs. 6 and 7 for higher magnification of IC and rostral LA. **e** Scheme showing bilateral injections of AAV1-DIO-ChR2:YFP into the VPMpc, light-evoked electrophysiological recording in IC and LA in *Cck^Cre* mice. **f** Recording electrode on top of IC and LA, scale bar 150 μm. **g** Sample traces and **h** summary figure of 470-nm blue-light pulse of light-evoked EPSCs in the IC ($n = 23$) and LA ($n = 17$) neurons of *Cck^Cre* mice. Sample traces (**i**, **l**), summary figures of TTX blocked EPSCs (**j**, **m**) and 4AP restated EPSCs (**k**, **n**) in IC ($n = 4$) and LA ($n = 4$) neurons of *Cck^Cre* mice. Data are represented as mean ± SEM.

of all DAPI-positive cells in VPMpc. Recent validated isotropic fractionator demonstrated the ratio of neurons to glia is close to 1:1[42,43]; therefore, the other 50% are likely to be glial cells. Thus, the *Cck/Oprm1* neurons represent most, if not all, of the VPMpc neurons; single-cell RNA sequencing may reveal cell-type heterogeneity of VPMpc neurons. Because of the co-expression, we used *Cck^Cre* and/or *Oprm1^Cre* mice in these studies.

Electrophysiological recording from either CCK^VPMpc or OPRM1^VPMpc neurons revealed that almost all of them responded to poststimulation of ChR2-expressing axon terminals from the CGRP^PBN neurons with a similar mean amplitude of ~150 pA. In retrospect, it is not surprising that CCK/OPRM1^VPMpc neurons respond to aversive stimuli because of the strong input from CGRP^PBN neurons that are known

to respond to most aversive stimuli. The glutamate signaling from the CGRP^PBN neurons is likely to be the most relevant transmitter, because expression of the receptors for some of neuropeptides (CGRP, NTS, PACAP and Substance P) made by CGRP^PBN neurons were not detected in the VPMpc. In contrast, neuropeptide signaling from CGRP^PBN neurons to the central nucleus of the amygdala is critical for fear learning and memory[44].

Inactivating the CCK/OPRM1^VPMpc neurons greatly attenuated the freezing response to foot shocks both during fear learning, and the memory of the context or cue 24 h later. These results resemble that achieved by inactivating CGRP^PBN neurons[23,25]. Optogenetic activation of *Calca* neurons can substitute for a foot shock in this fear-conditioning paradigm[23,25] and the photoactivation of *Calca* neuron

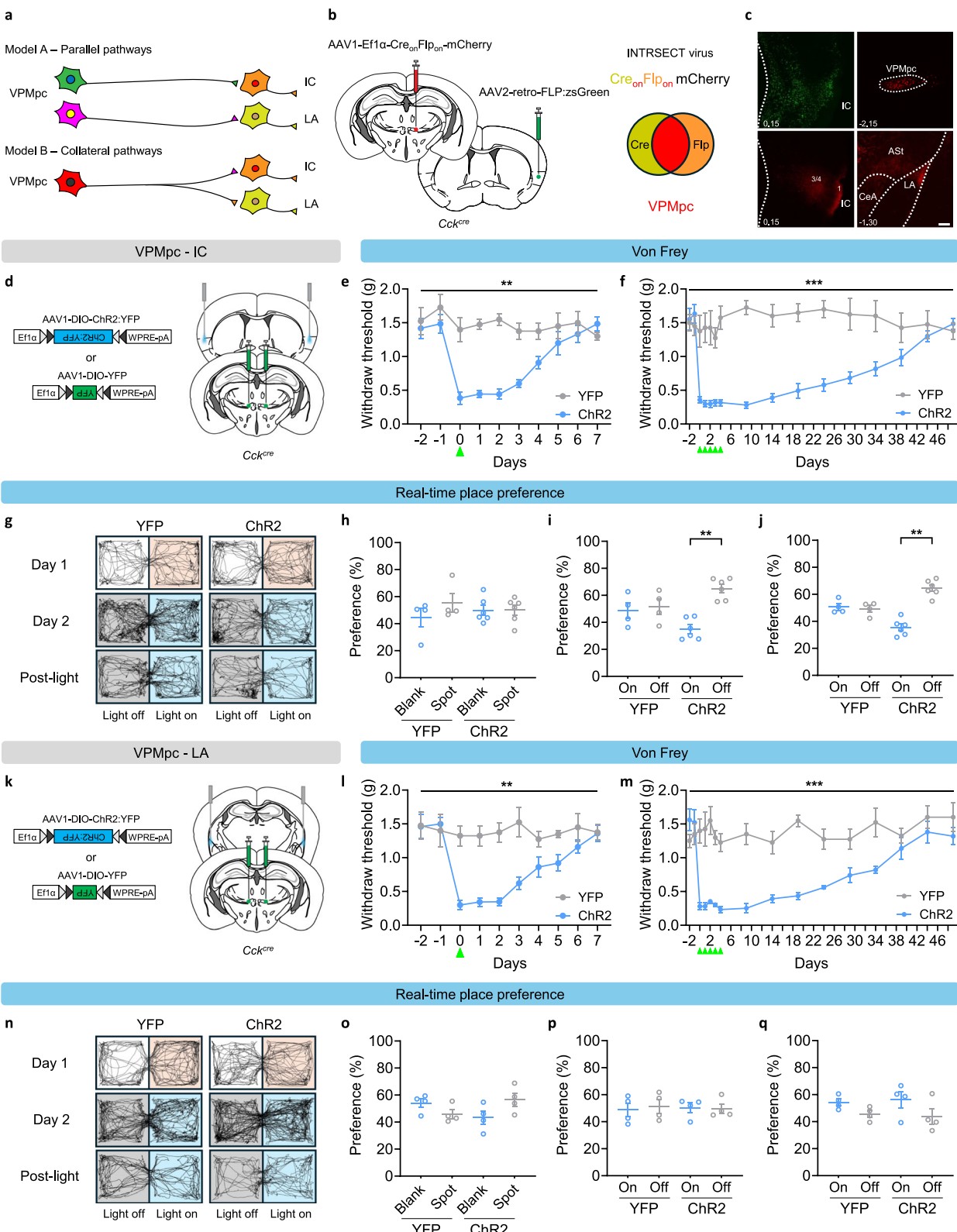

axonal projections in the VPMpc was able to substitute for foot shock[25]. However, directly activating the CCK/OPRM1[VPMpc] neurons was unable to serve as an unconditioned stimulus in the fear conditioning paradigm. The central tegmental fiber track from the PBN to the forebrain passes below the VPMpc[20,25]; thus, a likely explanation for this discrepancy is that photoactivation of CGRP[PBN] neuron terminals in the VPMpc also activated fibers of passage. This result emphasizes the

importance of being able to directly activate molecularly defined neurons in a brain region of interest.

Optogenetic activation of CCK[VPMpc] neurons resulted in elevated tactile sensitivity of the hind paws with similar kinetics to that observed when activating CGRP[PBN] neurons[45]. Five minutes of stimulation of either the *Calca* neurons in the PBN or *Cck* neurons in the VPMpc resulted in allodynia lasting a few days, whereas sequential

**Fig. 7 | VPMpc projection to IC encodes negative valence. a** VPMpc projection models. **b** Scheme showing retrograde injections of AAVretro-FLP:zsGreen into the IC and AAV1-Cre$_{on}$Flp$_{on}$:mCherry into the VPMpc. **c** Sample images showing zsGreen in IC and mCherry in VPMpc (top), VPMpc collateral projections in IC and LA (bottom), scale bar 150 μm. **d** Bilateral injections of AAV1-DIO-ChR2:YFP and AAV1-DIO-YFP into the VPMpc, optic fiber implantation above the IC. Von Frey test showing continuous days of decreased paw-withdraw threshold after 5-min (Two-way repeated measures ANOVA, F (1, 8) = 12.91, $P = 0.0071$) (**e**) and 5 days of 5-min (YFP: $n = 4$, ChR2: $n = 6$, two-way repeated measures ANOVA, F (1, 8) = 44.98, $P = 0.0002$) (**f**) photo-stimulation of VPMpc-IC terminals. **g** Representative mobility traces showing no side preference without photostimulation on Day 1, higher preference in light-off chamber in ChR2-expressing mice but not in YFP mice on Day 2 during and after light stimulation. **h–j** YFP mice ($n = 4$) and ChR2 mice ($n = 6$) had similar side preference without optical stimulation, avoidance of light-paired

side during (YFP: paired two-tailed $t$ test, $P = 0.8254$; ChR2: paired two-tailed $t$ test, $P = 0.0053$) and after optical stimulation (YFP: paired two-tailed $t$ test, $P = 0.8003$; ChR2: paired two-tailed $t$ test, $P = 0.002$). **k** Bilateral injections of AAV1-DIO-ChR2:YFP and AAV1-DIO-YFP into the VPMpc, optic fiber implantation above the LA. Von Frey test showing continuous days of decreased paw-withdraw threshold after 5-min (Two-way repeated measures ANOVA, F (1, 7) = 13.24, $P = 0.0083$) (**l**) and 5 days of 5-min (YFP: $n = 4$, ChR2: $n = 5$, two-way repeated measures ANOVA, F (1, 7) = 29.46, $P = 0.001$) (**m**) photo-stimulation of VPMpc-LA terminals. **n** Representative mobility traces showing no side preference in either YFP and ChR2 mice without photo-stimulation on Day 1, no preference in light-off chamber and light-on chamber on Day 2. **o–q** YFP mice ($n = 4$) and ChR2 mice ($n = 4$) had similar side preference without, during and after optical stimulation. Data are represented as mean ± SEM. *$P < 0.05$, **$P < 0.01$, ***$P < 0.001$.

activation for 5 days resulted in allodynia lasting several weeks. Silencing the activity of VPMpc neurons with TetTox resulted in a sustained analgesia in the von Frey, tactile-sensitivity test. These results suggest that VPMpc neurons are part of the neural circuitry downstream of the PBN that can promote nociplastic pain - defined as pain without nerve damage[45].

Norgren and Leonard reported in 1971 that an area in the pons now known as the PBN includes taste-responsive neurons[46] and they reported 2 years later that the PBN neurons relay taste signals to the gustatory thalamus, i.e., the VPMpc[13]. These results have been extensively confirmed in subsequent years[9–12]. The taste-responsive neurons in the PBN were primarily located in the waist and adjacent areas with some of them interspersed within the superior cerebellar peduncle fiber tract[12,47–49]. Neurons expressing the transcription factor *Satb2* reside in this region of the PBN, they respond to all tastants, and project to the VPMpc[21,50]. *Satb2*-expressing neurons in the PBN do not overlap with *Calca* neurons[20,50]. CGRP$^{PBN}$ neurons are activated by bitter tastes[51] and inactivating them attenuates responses to aversive (bitter and sour) tastes[50]. Our observation that most of the neurons in the VPMpc that express *Cck* and *Oprm1* are innervated by CGRP$^{PBN}$ neurons suggests that the synaptic inputs from *Satb2* and *Calca* neurons likely overlap in many VPMpc neurons. Deciphering how taste signaling by the VPMpc and the more general aversive signaling described here are distinguished in the VPMpc and downstream nuclei remains an intriguing puzzle.

VPMpc neurons have been shown respond to gustatory signals in both anesthetized and alert animals[6,8,16,18,52]. In an anesthetized rat study, VPMpc neurons also responded to thermal and tactile stimulation; applying distilled water at 0 and 37 °C into the open mouth and pinching tongue or palate by either a glass rod or non-serrated forceps could elicit VPMpc responses in anesthetized rats[8,16]. In our study, VPMpc neurons respond to tactile stimulation of the paws, exposure to a hot plate, appetitive and aversive taste stimulations. We also observed behavioral responses towards several other aversive stimuli, including freezing, withdrawal, escape and avoidance. Using single-cell analysis, we revealed that the same VPMpc neurons can respond to both taste and tactile stimulation. A few neurons had restricted responses to one modality whereas most of them responded to many modalities. More experiments are needed to enlarge the data set and ascertain if there is any pattern to the responsiveness of individual VPMpc neurons to various stimuli.

VPMpc neurons are well known to send ascending projections to the IC. A few early studies using anterograde and retrograde WGA-HRP methods observed connections from VPMpc to rostrodorsal part of the LA in cats and rats[53–55]. We observed that VPMpc project their axon terminals to medial to posterior IC and rLA. With slice electrophysiology, we also verified the functional connection between VPMpc and these two brain regions, with the response of IC neurons slightly stronger than that of the rLA neurons. In addition to IC and rLA, we also observed a weak VPMpc terminal signals in the CeA in agreement with

other studies[55,56]. The collateral experiments also showed VPMpc projections to CeA and ASt. Because of that, although the fiber-optic cannulas were placed over the rLA, we cannot rule out some activation of the neighboring CeA or ASt. The axonal projection of VPMpc neurons to the IC layers was prominent in layer 1 and layers 3-4. The synaptic properties and organization of VPMpc afferents in different IC layers has been investigated by others; layer 4 neurons had maximal light-evoked EPSCs from VPMpc but some neurons in all layers responded[57]. While the IC has a well-established role in gustatory signal processing, whether individual insular neurons respond exclusively to one tastant or multiple tastants is controversial[58–60]; our data obtained by recording individual VPMpc neurons favors the latter. The VPMpc and IC have reciprocal connections; it also receives inhibitory input from the reticular thalamic nucleus[11,57]. Application of contemporary retrograde tracing techniques may reveal more inputs to VPMpc neurons. The LA is important in fear memory formation and consolidation; synaptic plasticity in LA is necessary to associate the conditional stimulus and the unconditional stimulus during fear memory formation[61,62]. Thus, investigating how the projection from VPMpc to LA contributes to fear learning deserves examination. Together, our data expand the role of the VPMpc from a gustatory brain region to one that mediates a range of aversive behaviors.

## Methods
### Animals
All mice used in this study were backcrossed onto a C57BL/6J background for greater than 6 generations. Homozygous *Cck$^{Cre/Cre}$* mice were obtained from Jackson Laboratory (#012706). Homozygous *Oprm1$^{Cre/Cre}$* and *Calca$^{FLPo/FLPo}$* mice were generated and maintained as previously described[19,24]. *Calca$^{FLPo/+}$::Cck$^{Cre/+}$* and *Calca$^{FLPo/+}$:: Oprm1$^{Cre/+}$* mice were generated by breeding *Calca$^{FLPo/FLPo}$* with *Cck$^{Cre/Cre}$* or *Oprm1$^{Cre/Cre}$* mice, respectively. Heterozygous *Oprm1$^{Cre/+}$* were generated by breeding *Oprm1$^{Cre/Cre}$* with wide-type C57BL/6J mice. All mice were maintained on a 12-h light/dark cycle (7 am-7 pm) with food and water *ad libitum* in a temperature- and humidity-controlled animal facility at the University of Washington. Mice were group-housed (3–5 mice per cage) and single-housed after stereotaxic surgery. All experiments were performed during the light cycle and mice were randomized to experimental groups with both male and female mice from the same litter. No sex differences were noted. All animal experimental protocols were approved by the Institutional Animals Care and Use Committee at the University of Washington (Protocol #2183–02).

### Virus production
Plasmids for pAAV1-Ef1α-DIO-mCherry and pAAV1-hSyn-DIO-hM3Dq:mCherry were provided by B. Roth (Addgene #50462, #44361), pAAV1-Ef1α-DIO-YFP and pAAV1-Ef1a-DIO-ChR2:YFP were provided by K. Deisseroth (Addgene #27056, #35507). pAAV1-Ef1α-DIO-Synaptophysin:mCherry DNA plasmid was generated from pAAV1-Ef1α-DIO-

Synaptophysin:GFP with replacement of GFP[63]. AAV$_{DJ}$-Ef1a-DIO-GFP:TetTox, AAV1-CBA-DIO-GCaMP6m, AAVretro-CBA-FLP:zsGreen and the FLP recombinase-dependent AAV1-CBA-frtDIO-ChR2:YFP were generated by R. Palmiter. AAV1-Ef1α-Cre$_{on}$Flp$_{on}$:mCherry viral was generated by L. Zweifel at University of Washington. All viruses except for AAV$_{DJ}$-Ef1a-DIO-GFP:TetTox were prepared in-house by transfecting HEK cells with plasmids and pDG1 helper plasmids to coat AAV1 stereotype viruses. Viruses were then purified by mixing with sucrose and by CsCl-gradient ultracentrifugation. Viral pellets were re-suspended in 0.1 M phosphate-buffered saline (PBS) to yield approximately $10^{13}$ viral particles per mL. AAV$_{DJ}$-Ef1a-DIO-GFP:TetTox was prepared by the Janelia Viral Tools lab.

### Stereotaxic surgery

Mice were anesthetized with isoflurane (5% for induction, 2% for maintenance, flow rate 1 L/min) and placed on a robotic stereotaxic frame (Neurostar, Germany) for the entire surgery procedure. Mice also received local anesthesia of 1–2 mg/kg lidocaine and bupivacaine and skulls were surgically exposed. For calcium imaging recordings in vivo, a craniotomy was made unilaterally to target the VPMpc. AAV1-Ef1a-DIO-GCaMP6m virus (0.3 µl) was injected at a rate of 0.1 µl/min into the VPMpc (AP: −1.9 mm, ML: ±0.7 mm, DV: 4.1 mm) and a 0.6 × 7.3 mm Integrated ProView GRIN lens (Inscopix, #1050-004413, California, USA) was placed above the virus injection target. For in vivo optogenetic behavior recordings, 0.3 µl AAV1-Ef1a-DIO-ChR2:YFP virus was unilaterally injected into the VPMpc with the same coordinates and an optic cannula was placed 300 µm above the virus injection sites. Super glue (Loctite, Ohio, USA), dental cement (Lang Dental, Illinois, USA) and C&B metabond (Parkell, New York, USA) were used to secure both GRIN lens and optic cannulas. Other virus injection experiments were either targeted to the VPMpc with the same coordinates or PBN (AP: −4.9 mm, ML: ± 1.35 mm, DV: 3.4 mm) with a volume of 0.3 µl to VPMpc and 0.5 µl to PBN at a rate of 0.1 µl/min for 2 min using a Hamilton syringe (Reno, Nevada, USA). For Nanoject viral injection surgeries, mice were placed on a stereotaxic frame (David Kopf Instruments, California, USA) and a 15 nl viral mix of AAV1-Ef1a-DIO-ChR2:YFP and AAV1-Ef1α-DIO-Synaptophysin:mCherry was unilaterally injected into VPMpc with the same coordinates. We waited 10 min before withdrawing the injection needle. Following surgery procedures, 5 mg/kg Ketoprofen for analgesia was subcutaneously injected into mice which were monitored until complete recovery from anesthesia. All mice had a minimum 4 weeks of recovery before the start of behavioral experiments. Histological verification of viral injections and GRIN lens/ fiber optic placements were performed at the end of each experiment.

### Electrophysiology

**Slice preparation.** All mice were anesthetized with Euthasol (1 ml per 10 g body weight, i.p.) and intracardially perfused with 95% O$_2$/5% CO$_2$ saturated ice-cold cutting solution containing (in mM): 92 N-methyl-D-glucamine, 25 D-glucose, 2.5 KCl, 10 MgSO$_4$, 1.25 NaH$_2$PO$_4$, 30 NaHCO$_3$, 0.5 CaCl$_2$, 20 HEPES, 2 thiourea, 5 Na-ascorbate, 3 Na-pyruvate. After the intracardial perfusion, the mice were decapitated and the brain was removed and stored in the same ice-cold cutting solution superfused with 95% O$_2$/5% CO$_2$ saturated ACSF. Coronal slices (300 µm) were prepared using a vibratome (Leica VT1200, Illinois, USA) in the same ice-cold cutting solution. Brain slices were collected into 33 °C cutting solution for 10 min and then transferred to a room temperature recovery solution containing (in mM): 13 D-glucose, 124 NaCl, 2.5 KCl, 2 MgSO$_4$, 1.25 NaH$_2$PO$_4$, 24 NaHCO$_3$, 2 CaCl$_2$, 5 HEPES for at least 1 h. Brain slices were then individually transferred to a 33 °C artificial cerebral spinal fluid (ACSF) containing (in mM): 11 D-glucose, 126 NaCl, 2.5 KCl, 1.2 NaH$_2$PO$_4$, 26 NaHCO$_3$, 2.4 CaCl$_2$, 1.2 MgCl$_2$ for whole-cell patch recordings. All cutting, recovery, and ACSF solutions were saturated with 95% O$_2$/5% CO$_2$ and adjusted to pH 7.3–7.4, 300–310 mOsm.

**Data acquisition and analysis.** All recordings were obtained with glass pipettes (3–5 MΩ) filled with either Cs$^+$ or K$^+$ intracellular solutions. Cell-attached measurements were used to validate the efficiency of CNO applications using K$^+$ intracellular solutions containing (in mM): 135 K-gluconate, 4 KCl, 10 HEPES, 4 Mg-ATP, 0.3 NA-GTP (pH 7.35, 280−300 mOsm). Action potentials were recorded in voltage clamp with 0-pA holding current before and after bath applied 3 µM CNO onto thalamus slices with hM3Dq expression. To verify the connectivity between innervated neurons, light-evoked post-synaptic currents were recorded with Cs+ intracellular solutions containing (in mM): 120 CsMeSO$_3$, 20 HEPES, 0.4 EGTA, 2.8 NaCl, 2.5 Mg-ATP, 0.25 Na-GTP, 5 QX-314 bromide (pH 7.35, 280−300 mOsm). For PBN-VPMpc connectivity verification, whole-cell patch-champ was performed on mCherry-positive VPMpc neurons after identifying the YFP expression in the PBN cell body and seeing dense green fibers in VPMpc with epifluorescence microscopy (OLYMPUS BX51WI, Evident, Massachusetts, USA). For measuring VPMpc-IC and VPMpc-LA connectivity, only neurons with dense YFP fibers from VPMpc were patched. All neurons were recorded in voltage clamp with −70 mV holding potential and the post-synaptic currents were evoked by a 0.1-Hz 10-ms blue light pulsed delivered through the 40× objective via a SOLIS-3C high power LED (Thorlabs, New Jersey, USA) and filter cube (Chroma technology, Vermont, USA). Averaged data were obtained from 10 sweeps. All data were obtained using MultiClamp 700B amplifier (Molecular Devices, California, USA). Data acquisition and analysis were done using pClamp 11 and Clampfit 11.3 (Molecular Devices, California, USA).

**Optogenetic stimulation.** After allowing 6 weeks to recover from viral surgery and optic cannula implantation, dummy cables were attached to acclimate mice prior to testing. For behavioral studies, mono 200-µm diameter fiber-optic cables (Doric Lenses, Québec, Canada) were attached to the optic cannula of each mouse before experiments. A blue-light laser (473-nm; LaserGlow, Ontario, Canada) was used to stimulate ChR2 (10-ms light-pulse trains were delivered at 10 Hz, 2 ± 0.5 mW) using a Master-8 pulse stimulator (AMPI, Maryland, USA) in all behavioral tests. For von Frey-testing using optogenetic stimulation, 5 min of light stimulation (as above) was delivered in home cages 45 min before the tactile stimulation tests.

### Calcium imaging

Mice were prepared and recorded as previously described[45,64]. After allowing 6 weeks to recover from GCaMP6m virus injection and GRIN lens implantation surgery, each mouse's head was attached to a micro endoscope (Inscopix, California, USA) and connected to nVista 3.0 (Inscopix, California, USA) to acquire calcium images. Ethovision XT15 (Noldus, Virginia, USA), Med Associates control panel (Med Associates, Vermont, USA) and Clampfit 11.3 (Molecular Devices, California, USA) were connected to nVista 3.0 via BNC cables to trigger TTL pulses to allow synchronizing calcium signals with behavioral video recordings and conditional and unconditional stimuli. All mice recorded were freely moving and without any anesthetization. A commutator (Inscopix, California, USA) was used to avoid the overspinning of the micro endoscope during recording.

For sensory stimuli, mice were placed in their home cage and went through multiple trials of a transient 90 dB hand clapping sound, air puff on the face, tail pinch, and 10 s of lifting the mouse by the tail 0.5 m within one recording session, with a random ITI of 60–120 s. Heatmaps of sensory stimuli used the first trials for each mouse. For foot shock only recordings, mice were placed in the shock chamber and allowed 5 min free exploration before shock session started. During the shock session, mice went through 1–2 trials of 0.15, 0.3, 0.5 mA foot shock stimulation with an averaged ITI of 120 s. For fear conditioning recordings, mice were placed in the test chamber 5 min before calcium imaging starts across all three days. On Day 1, 10 trials of 10 s of 5 kHz, 65 dB tone were delivered with a 100-s ITI. On Day 2, 10

trials of 10 s, 5 kHz, 65 dB tone were co-terminated with 0.5-s, 0.5-mA foot shocks (ITI 100 s). On Day 3, mice were placed in a different chamber and only 10 trials of 10 s, 5 kHz, 65 dB tone were delivered with an ITI of 100 s.

For taste stimuli, mice were placed in a custom lickometer cage[65] with sipper tube water ports opening on the front wall. Cage floor was covered with aluminum foil and a wire-grid. The stainless-steel sipper tube on taste bottles was connected to an analog-digital converter (Digidata 1440A, Molecular Devices, California, USA) to record the number of licks. Mice were acclimated to the lickometer cage with water bottle held in the cage wall for minimum 3 day and water restricted the day before test day. During taste exposure, mice were given 5 s access to each taste solution during each trial starting from the first lick of the sipper tube. Water, 5% sucrose and 0.1 mM quinine were presented to mice with a random ITI of 60–120 s. The licks were recorded with a video camera and synchronized to nVista 3.0 using a TTL trigger.

Calcium fluorescence was recorded at 20 frames per second and under LED light while mice were being tested. The recording parameters were chosen from pilot recordings with minimal photobleaching but sufficient detection of fluorescence. Raw calcium recording videos were pre-processed with Inscopix Data Processing Software (IDPS 1.9.1, Inscopix, California, USA) with 2× down sampling of frame and spatial bandpass filter to reduce processing time and background noise. Rigid motion correction algorithm provided by IDPS was also applied to the filtered images to minimize motion artifact during analysis. Constrained non-negative matrix factorization for microendoscope data (CNMF-E) analysis was used to extract the raw F over noise of each neuron within the region of interest in field of view[66]. All CNMF-E detected calcium transients were visually inspected for each cell to ensure accuracy. Raw traces (F over noise, equivalent to F) after CNMF-E were analyzed using custom code in Matlab to generate $\Delta F/F$ and Z score. $\Delta F/F$ was calculated as $\Delta F/F = ([F-mean(F0)])/(mean(F0))$. Z score was calculated as $Z = ([F-mean(F0)])/(Standard\ deviation(F0))$. F indicates the fluorescence at any given point and F0 indicates the average fluorescence of −20 to 0 s in fear conditioning recordings and −10 to 0 s in all other recordings.

The area under the curve (AUC) of $\Delta F/F$ were used to identify whether the cell was activated, inhibited or unresponsive. AUC was calculated for both pre-stimulation (−10 s to 0 s) and post-stimulation (0 s to 10 s) across all trials. Neurons were deemed as 'increased' if the AUC of post-stimulation has >50% increase compared to the AUC of pre-stimulation. Neurons were deemed as 'decreased' if <50% and redeemed as 'no response' if the difference of AUC did not show >50% increase and decrease.

## Behavior measurements

**Open-field test.** Mice injected with YFP or ChR2 were attached to fiber-optic patch cords and placed in a 40 cm × 40 cm × 30 cm white plexiglass open field box for at least five minutes of habituation. Photostimulation was for 30 s (473 nm, 10 Hz and 2 mW) and delivered 3 times with one-minute intervals between stimulations. Mice were attached with fiber-optic patch cords for both control and experimental groups to minimize the variation. The distance traveled was collected and analyzed using video-tracking software EthoVision XT 15 (Noldus, Virginia, USA).

**Von Frey test.** The von Frey tactile sensitivity test was performed as described[45]. Mice were placed in a 30-cm elevated metal mesh floor with 11.5 cm × 7.5 cm grid separated by black plexiglass (Bioseb, Florida, USA) for at least 30 min before experiments. Von Frey filaments (Bioseb, Florida, USA) with different stiffness ranging from 0.07 to 4 g were used to measure the tactile sensitivity of the hind paw plantar surface in each mouse. Filaments were applied from the minimum weight and switched to a higher weight if there were less than 3 paw-withdrawal responses within 5 trials. The lowest weight that elicited 3 paw-withdrawal responses with 5 trials was recorded as paw withdraw threshold and confirmed with the next consecutive filament. The ipsilateral and contralateral paw-withdrawal thresholds were separately measured and combined in the data analysis due to no differences observed in left/right paw-withdrawal thresholds.

**Hot-plate test.** A hot-plate test was conducted as described[45]. Mice were gently placed on a 52.5 ± 1 °C pre-heated 16.5 × 16.5 cm aluminum surface embedded in a 16.5 × 16.5 × 35 cm rectangular clear plexiglass enclosure (Bioseb, Florida, USA). The latency to the first jump and the number of jumps were recorded and measured in 60-s trial tests. Additional time (maximum 30 s) were given to mice resisted to jump in the 60-s trial until it jumped. For individuals that resisted to jump, the animal was immediately removed from the hot plate after 90 s to avoid tissue damage.

**Fear conditioning.** Mice were placed in a fear-conditioning chamber consisting of a 20 × 14 × 13 cm shuttle box with a metal grid floor, a light on the wall and a speaker in the back (Med Associates, Vermont, USA). The metal grid floor is made of stainless-steel bars and delivers an electric shock. The metal grid floor, light and speaker were all connected and controlled by Med Associates software. A 3-day series of trials were conducted. Day 1: Mice were placed in the chamber and allowed free exploration for 60 s before the onset of a discrete conditioned stimulus, which consisted of 6 trials with a 20 s, 5 kHz, 65 dB tone with a 100-s inter-trial interval (ITI). Day 2: Mice were allowed to explore the apparatus for 60 s and 6 trials of 20 s, 5 kHz, 65 dB CS tone was delivered that co-terminated by a 2-s, 0.5-mA foot shock with an ITI of 100 s. After the sixth CS-US paring trial, mice remained in the chamber for another 100 s before returning to the home cage. Day 3: For the contextual test, mice were placed in the same chambers for 2 min of free exploration. For the cue test, mice were placed in a different chamber and allowed to explore the cage for 60 s and then one CS tone (20 s, 5 kHz, 65 dB) was delivered. Freezing data were recorded in a camera synchronized with the Med Associates software and freezing time (immobility time with both mouse head and body) during each trial was manually scored by the examiner.

**Fear-potentiated startle.** This test was performed as described[67]. Mice were placed in the standardized enclosure of a sound-attenuating startle reflex chamber (SR-LAB, San Diego Instruments, California, USA) for 5 min of free exploration with a background of 65 dB white noise. 7 trials of escalating sound from 65 (null), 80, 90, 100, 105, 110 and 120 dB were presented to mice with an ITI of 30 s. This 7-trial series was repeated 10 times with the same ITI. All sounds were delivered with 40-ms pulses. Startle responses upon each sound level changer were calculated and averaged for all 10 repeated trials. For fear-potentiated startle, mice undergo a 3-day protocol. On Day 1 (baseline day), mice were pseudo-randomly given 10 trials of cue and 10 trials of no-cue conditions after a 5-min habituation period. For cue trials, mice were presented with 10-s light and a 40-ms, 105-dB sound pulse co-terminating with the light. For no-cue trails, mice were only presented with 40-ms, 105-dB sound pulse without any light cue exposure. The ITI was 60–180 s with an average of 120 s. On Day 2 (training day), after a 10-min habituation period in the startle chamber, mice were given 30 trials of 10-s light cue stimulation, co-terminated with 0.2 mA, 0.5 s foot shock. The ITI between these 30 trials also ranged from 60 to 180 s, with an average of 120 s. On Day 3 (test day), mice underwent the same protocol as on Day 1. All no-cue and cue trial responses were averaged separately on Day 1 and Day 3. Fear-potentiated startle was calculated as [(averaged responses on cue trials/average responses on no-cue trials − 1) ×100].

**Conditioned place preference (CPP).** Mice underwent a 4-day conditioned place preference paradigm protocol in a rectangular CPP box. The CPP box is divided equally into two 20 × 20 × 30 cm square boxes with a 6.5 × 30 cm slide door connected in the middle. These two square boxes differ in wall decoration with vertical stripes one side and spots on the other side. The vertical striping chamber also contains two triangular blocks in the corner. On day 1, Mice were placed in the middle of the CPP chamber and allowed free exploration for 30 min with open access to both striped and dotted chambers. On day 2 and day 3, mice were i.p. injected with Saline (10 ml/kg) and immediately moved to the saline-paired chamber (either the vertical stripe or circled dot chamber) with the door closed. Mice were allowed free exploration in the saline-paired chamber for 30 min. Five hours later, mice received the same volume of U50 (5 mg/kg) through i.p. injection and were moved immediately to the other chamber paired with U50. Mice were allowed free exploration in the U50-paired chamber for 30 min and returned to their home cage after. On day 4, mice were placed in the middle of two chambers and allowed to freely explore both sides for 10 min. The time spent on each side was measured and averaged using the video-tracking software, EthoVision XT 15 (Noldus, Virginia, USA), to calculate the conditioned preference.

**Real-time, place preference (RTPP).** RTPP test was performed as previously described[25]. The RTPP box is a rectangular box divided into two equal-sized square box (60 × 30 × 30 cm) with one side painted with black dots on the wall and the other side being blank without any symbols. The dotted and blank chambers were connected through a single doorway (10 × 30 cm) in the middle. Mice injected with YFP or ChR2 were attached to fiber-optic patch cords and underwent two days of RTPP test paradigm. On day 1, mice were placed in the middle of the two chambers and allowed free exploration for 20 min without any light stimulation. On day 2, mice were also placed in the middle of the chamber and allowed to be freely moving for 20 min, but one side of the chamber was paired with optical stimulation. Mice received 473 nm, 10 Hz, 2 mW light stimulation once it moved to the stimulation chamber and the light stimulation was continuously turned on until it moved out of the light-paired chamber. The time spent on the light-paired chamber and the no-light chamber was measured and averaged using the video-tracking software, EthoVision XT 15 (Noldus, Virginia, USA), to calculate the real-time place preference.

### RNAscope in situ hybridization
Mice injected with AAV expressing hM3Dq were given CNO (1 mg/kg, i.p.) and put back in their home cage. After 60 min, mice were anesthetized with Euthasol (1 ml per 10 g body weight) and decapitated. Brains were quickly removed and rapidly frozen on crushed Dry Ice. coronal cryostat sections (20 μm) were cut on cryostat (Leica Biosystems, Illinois, USA) and directly transferred to a SuperFrost Plus glass slide (Thermo Fisher Scientific, Massachusetts, USA) and frozen at −80 °C. RNAscope fluorescent multiplex V1 assay was performed following the manufacturer's protocols (ACD, California, USA) using the following probes: *Cck*, *Oprm1* and *Fos* (ACD, California, USA). All images were collected on Keyence BZ-710 microscope (Keyence, Illinois, USA) and Olympus FV-1200 confocal microscope (Evident, Massachusetts, USA). RNA expression and colocalization were analyzed using Fiji with manually drew regions of interest. Cells with at least 4 puncta associated with a DAPI nucleus expression were considered positive.

### Histology and immunohistochemistry
The detailed procedures for mice brain slice preparation were described[20,45,68]. Mice were anesthetized using pentobarbital (0.2 ml, i.p.) and transcranial perfused with 1× ice-cold phosphate-buffered saline (PBS) followed by 4% ice-cold paraformaldehyde (PFA, Electron

Microscopy Sciences, Pennsylvania, USA) made in PBS. For virus injection verification, each brain was rapidly removed and fixed overnight in 4% PFA at 4 °C. For optic cannula and lens implantations, the whole mouse head was decapitated and moved to 4% PFA at 4 °C for at least 24 h and the brain was then dissected and post-fixed in 4% PFA at 4 °C for 6 h. Then the brain was transferred to 30% sucrose for 24–36 h until it was fully saturated and sank. The brain was frozen in OCT compound (Thermo Fisher Scientific, Massachusetts, USA) and stored at −80 °C. Before sectioning, the brain was transferred from −80 °C to −20 °C for at least 30 min to prepare for the cryostat sectioning (Leica Biosystems, Illinois, USA). Coronal cryostat sections (35 μm) were cut on cryostat and collected in ice-cold PBS.

For immunohistochemistry, sections were washed 3 times in 1× PBS and blocked with 3% normal donkey serum made in PBST (0.2% Triton X-100 in PBS) for 1 h at room temperature. Sections were then incubated overnight at 4 °C in a blocking solution with primary antibodies including chicken-anti-GFP (1:10000, Abcam, ab 13970) or rabbit-anti-dsRed (1:2000, Takara, Cat# 632496). After washing the residual primary antibodies with PBS for 3 min the following day, sections were incubated at room temperature for an hour in PBS with secondary antibodies including Alexa Fluor 488 donkey anti-chicken (1:500, Jackson ImmunoResearch, #2340375) and Alexa Fluor 594 donkey anti-rabbit (1:500, Jackson ImmunoResearch, #2340621). Following washing with PBS for 3 times, the sections were mounted onto SuperFrost Plus glass slides (Thermo Fisher Scientific, Massachusetts, USA) and coverslipped with Fluoromount-G with DAPI (Southern Biotech, Alabama, USA). Fluorescent images were collected on Keyence BZ-710 microscope (Keyence, Illinois, USA) and Olympus FV-1200 confocal microscope (Evident, Massachusetts, USA). Images were minimally processed using Fiji to enhance brightness and contrast. All images were processed the same way.

### Statistical analyses
All the average data are shown as mean ± standard error of the mean (SEM). Shapiro–Wilk Test was performed to determine whether parametric tests could be used. All data were evaluated by unpaired two-tailed *t*-tests, paired two-tailed *t*-test, Mann–Whitney test, Wilcoxon signed rank test, two-way ANOVA, two-way repeated measures ANOVA, followed by Holm-Sidak's multiple comparisons when appropriate. All statistical tests were indicated where used. $P < 0.05$ was considered as statistically significant and corresponding to the following values: *$P < 0.05$, **$P < 0.01$, ***$P < 0.001$. Independent sample size (n) represented the number of animals in behavioral tests, the number of brain slices in RNAscope analysis, and number of neurons in electrophysiological recordings. Mice were excluded from experimental analysis if the viral expression was inadequate, or off-target based on histology and imaging verification. All analyses were conducted using GraphPad PRISM 10 (GraphPad Software, California, USA) and MATLAB R2023b (MathWorks, Massachusetts, USA). The statistical analysis for each individual experiment is summarized in Supplementary Table 1.

### Reporting summary
Further information on research design is available in the Nature Portfolio Reporting Summary linked to this article.

## Data availability
Source data are provided with this manuscript. Other Data are available from the corresponding author upon request. Source data are provided with this paper.

## Code availability
The code used in this study is available from Zenodo (https://zenodo.org/records/16609026).

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

## Acknowledgements
We thank Susan Phelps and Lucy Anastas for maintaining the mouse colony, Larry Zweifel and his lab members for AAV preparation and insightful discussions, Anqi Zhu for technical assistance with MATLAB analysis and members in Palmiter lab for comments on the manuscript.

## Author contributions
F.C. and R.D.P. conceived and designed the study. F.C. and S.P. performed and analyzed the 1-photo calcium imaging. F.C. performed and analyzed the electrophysiology experiments. F.C., E.Y.S. and A.N. performed and analyzed the behavioral experiments. F.C., J.L.P., E.Y.S. and A.N. performed and analyzed the histology experiments. F.C. wrote the manuscript with input from all authors.

## Competing interests
The authors declare no competing interests.
