## [Transparent Peer Review file · Nature Communications]

Gustatory Thalamic Neurons Mediate Aversive Behaviors

Corresponding Author: Dr Richard Palmiter

Version 0:

Reviewer comments:

Reviewer #1

(Remarks to the Author)

In this study, Cao et al provide series of evidence that the gustatory thalamus (VPMpc) has the ability to mediate aversive behavior. Using single cell Ca-imaging they show that VPMpc cells expressing CCK respond to thermal, tactile and electrical painful stimuli as well as to learned auditory signals. They also show that these cells can bidirectional control various forms of aversive behaviors.

The study has several strengths though also some limitations. Overall, the authors provide a valuable contribution to the field by expanding the list of neuronal mechanism for aversive behavioral responses with the gustatory thalamus.

My general comments and questions.

1) It was stated that “Thus, the Cck/Oprm1 neurons represent most, if not all, of the VPMpc neurons;...” . However, can it be really ruled out that there is a CCK-negative neuronal population which belongs to the gustatory thalamus? Going further on this line, earlier studies – also cited by the authors, like Yasui et al., 1987 – showed that VPMpc also projects to the central amygdala. Recently, rabies-mediated tracing also showed prominent VPMpc-CeA connection (Douglass et al., 2017). According to the figure 6 and the extended fig 7, CCK cells do not project to the CeA. These could suggest that there is an other (non-CCK) VPMpc cell group which targets CeA, and so, the CCK expression territory is only a subfield of the gustatory thalamus. One solution could be to examine the CCK-content of the insular cortical-projecting VPMpc cells. A large or total overlap between retrogradely labeled cells and CCK could provide a firm argument for investigating VPMpc based on the CCK expression.

Of course, even if there is an other VPMpc population, the main conclusion of this study is still valid (if the neuronal perturbations using CCK-Cre line and Cre-dependent viral constructs were limited to VPMpc; see next comment) but only for the CCK population. Then, the title and the text should specify this.

2) This is a methodically elegant study combining various behavioral tests, calcium imaging and in vitro electrophysiology with viral tracing. In Figure 6, the authors showed the results of a 15 nl AAV injection. Indeed, selective targeting of a small region like VPMpc requires that precision. Furthermore, the AAV signal amplification with antibody staining was also performed which is highly welcomed. However, in the rest of the experiments, 300 nL of AAVs were injected in the thalamus. In the view of the extended expression of CCK in thalamic areas (even in the neighborhood of VPMpc, including PIL/SPFp or posterior PVT), it is likely that this large volume of AAV transduced not only the VPMpc but also other thalamic territories. It would be necessary to show the size and regional distribution of the AAV infections (with antibody amplification) for each experiment.

3) Based on the findings it is clear that VPMpc can also serve as US (or US-associated) signal for LA-dependent fear learning. However, it is also stated that “However, directly activating the CCKVPMpc neurons was unable to serve as an unconditioned stimulus in the fear conditioning paradigm (F.C., unpublished).” Is it not contradiction?

It is interesting that VPMpc cells only projects to the rostral part of the LA. Does it mean that rest of the LA receives US-related signals from other sources? Indeed, certain MGN regions and neighboring thalamic areas (like PIL/SPFp) has been shown to send associative/multisensory aversive signal rather than pure US to the LA (Barsy et al 2020, Taylor et al, 2021, Kang et al., 2022) without any topography. Barsy et al (2020) showed that a calcium binding protein, the calretinin (CR) marks most of the thalamic territories including gustatory thalamus which send projection to the LA. Furthermore, CR thalamic cells also heavily project to layer 1 of insular and other associative cortical areas similar to VPMpc. Thus, VPMpc shares many similarities with PIL/SPFp.

Thus, a question could arise why distinct thalamic regions projects to LA. Do they code slightly distinct elements (features) of the environment and provide these as associative signals (feature1-US, feature2-US, ect) to the LA? Some sort of

association can already take place before LA, at thalamic or even upstream to it, at the superior colliculus or parabrachial nucleus level.

Other comments and questions:

- a) Most of avg Ca transients start to elevate before the onset (Fig 2). What is the explanation for that?
- b) Fig 2 and 3. The avg Z score values are between 1 and 2, except in the cases of foot shock trials. Still, the scaling of the heat maps is from -6 to 6 everywhere (or even larger in the extended figures), except in the US panels, where it is -4 and 4. Why is this different scaling? For valid comparisons, the same scaling would be preferred.
- 4) The n numbers for imaged cells (in Fig 2-3) are varying from one experiment to other although these were collected from the same animals. What is the reason for that?
- 5) Fig 2 and 3. In what order are the cells placed on the heat map. Were those always placed in the same order?
- 6) As all of these experiments are performed using the same animals, is it possible to track the same cell throughout the all experiments? This type of analysis could reveal whether the same cell can code various forms of aversive signals or there are stimulus specific groups of CCK VPMpc cells.
Furthermore, tracking the same cell throughout the 3 days of fear-learning paradigm, could show the presence of any plastic changes of thalamic cells previously shown in other, amygdala projecting thalamic population (Taylor et al, 2021, Bارسy et al 2020).
- 7) Fig 5c. Either the y axis title or the scaling is not correct.
- 8) Is there any data, either in the literature or in the authors' hand, showing the projection-selectivity of the CCK VPMpc cell? Whether the IC and LA projecting cells are the same or belong to distinct VPMpc subgroups? If the CCK VPMpc cells have convergent projection to IC and LA, then, the observed behavioral changes (with TetTox, hM3Dq, ChR2) can be due to either circuit. Otherwise, as it is mentioned in the discussion, projection-selective perturbation is required to reveal the existence of functional specificity of the VPMpc-LA and VPMpc-IC pathways.

Reviewer #2

(Remarks to the Author)

Does activation/inhibition of these neurons induce baseline changes in locomotion? The open field experiments suggest that activation may increase locomotion and inhibition may decrease locomotion, which could have significance in interpreting many of the other outcomes suggested to be a sign of aversion.

For recording experiments in VPMpc neurons: in almost all cases the effects on activity are pretty subtle and there are as many neurons that decrease their activity as increase, yet this is not mentioned or discussed.

How this pattern of activity (some decrease and some increase) maps onto the gain and loss of function experiments could be discussed too.

Do salient but non-aversive experiences increase activity of these neurons?

Several of these interventions would induce activity in brain areas that would not be classified as mediating aversion (clapping induces calcium signals in V1 neurons for example). The fear learning result ameliorates this a little, but are the authors sure that the neural responses are due to aversion rather than surprise or startle?

As the stimuli provided are all likely to induce some kind of startle and/or motion are the authors confident that the increases/decreases are not due to a shift in the imaging z-plane.

Figure 2 j - this plot is incorrect (values don't match size of shaded area)

Minor comments:

-In Figure 2 heatmaps. The colorbar axes are too broad to fully appreciate the activity changes in most neurons. I could understand if the authors were trying to fix the axes of all heatmaps but they do differ in c vs p for example.

-In Figure 2 b,e,h,k - is this the average of all recorded neurons? Text in figure legend is not clear.

Reviewer #3

(Remarks to the Author)

This manuscript titled "Gustatory Thalamic Neurons Mediate Aversive Behaviors" examines the role of thalamic subnuclei circuitry in fear learning and threat behavior. The VPMpc is generally thought to be involved in taste aversion, but the authors argue it may play a more prominent role in aversive behaviors. The manuscript elegantly demonstrates that Cholecystokinin (Cck) and mu-opioid receptors (Oprm1) are co-expressed in the same VPMpc neurons, and these neurons can be manipulated to alter aversive behaviors. The manuscript is well-written, and the experiments are technically innovative. However, there are several weaknesses and concerns that detract from my enthusiasm. The hypothesis is that the VPMpc transmits aversive signals to the IC to mediate aversive behaviors, but this isn't directly tested. The advantage and significance of using Cck and Oprm1 as molecular markers over examining the PBN-VPMpc-IC circuit independent of molecular markers is unclear. For example, is the goal to target Cck ultimately pharmacologically and/or Oprm1? This concern is compounded by the inconsistent use of Cck-Cre and Oprm1-Cre mice throughout the manuscript. In addition, I have larger concerns about the lack of statistical rigor, as presented. In all, the manuscript has the potential to have a high impact given the novelty of the VPMpc playing a role in non-taste-related aversive behavior. Still, this manuscript is somewhat disjointed at times and would benefit from a more focused narrative and better-outlined goals and conclusions.

Specific comments are summarized below:

1)The justification for interchangeably using both mouse lines is lacking. Some experiments use all of one strain, and some use a mix, which is a concern. It seems like focusing on one mouse line would be more efficient from a narrative perspective. Examples:

- a.Line 85-88: "Fos was expressed in >18% of either Cck- or Oprm1-expressing neurons..." but the figure legend for 1h says the data is just Cck neurons.
- b.Figure 4: A-D is CCK-Cre, E-J is a mix, and K-M is Oprm1-Cre.

2)The authors argue that the Cck/Oprm1 expressing neurons represent "most, if not all of the VPMpc neurons" (Line 269). I'm not grasping the significance of these molecular markers if essentially the entire VPMpc (at least the neurons) is represented.

3)If I understand the data correctly, fos was expressed in 18% of Cck+ neurons after CGRP chemogenetic activation. As this is a marker of increased neuronal activity, it fits nicely with the mini-scope data showing a similar percentage of cells that increase during the behavior. But is fos expressed in neurons that do not express Cck? What is the significance of the 82% of Cck neurons that do not express fos? This experiment could generally use more detailed analysis and discussion of results and interpretation.

4)Figure 1E shows data points with over 100% colocalization. I'm not sure how this is possible.

5)Figure 2 shows bulk fluorescence increases, but the percentage of cells increasing is small. Are these the same cells? From the heatmaps, it looks like generally no.

- a. Are the percentages similar between animals?
- b. Was foot shock performed in separate mice?
- c. Why does the number of cells differ between tests? The n is 76 for startle, 29 for FS, and then 59 for FC in figure 3.

6) Lack of statistical comparison. Line 148- "the response was much greater than the response on day 1". The area under the curve during and/or after the tone should be measured to make this conclusion.

7)In general, there needs to be more statistical rigor throughout. Many tests are just reported as two-way ANOVAs with very little detail. Effects should be reported for interactions, group, treatment, post-hoc comparisons, etc. For example, the CPP in Figure 4m is within-subject, so the data is represented twice (70% on the saline side is the same as 30% on the U50 side). The only comparison should be YFP vs. TetTox on either the saline or U50 sides.

8)The data in Figures 2 and 3 suggest a shift from neurons that decrease activity to neurons that increase activity. This is an exciting finding, but Figure 4 broadly inhibits all Cck/Oprm neurons, not just the percentage of neurons that increase activity. How do these findings relate? More discussion comparing results across figures is needed.

9)The authors identify projections from the VPMpc to the insula, but this is already an established bidirectional pathway. A functional role for this pathway in aversive behaviors would strengthen the manuscript.

10)Similarly, electrophysiology establishes the functional connection between CGRP neurons and Cck/Oprm1 in the VPMpc. Still, the author does not present data demonstrating that activating CGRP terminals in the VPMpc affects aversive behavior.

11)Do all VPMpc neurons project to the LA and the IC (collaterals)? Or do subpopulations of neurons project to either the IC or LA? This distinction is essential, considering the tet experiment targets all Cck or Oprm1 neurons. Retrograde labeling from the IC and LA and assessing colocalization in the VPMpc would address this.

12)Lines 287-288 show that directly activating CCK neurons was unable to serve as an unconditioned stimulus to fear conditioning. This seems like exciting data that should be included.

13)Minor- Extended Figure 1 mentions Crhr1-Cre, but these data are not shown.

14)Minor- Lines 311-314 referencing data "to be published elsewhere" should be removed as the data cannot be verified.

Reviewer #4

(Remarks to the Author)

Cao et al., identified neurons expressing cholecystinin (CCK) and mu opioid receptor (Oprm1) in the parvicellular part of the ventral postmedial nucleus (VPMpc) of the thalamus transmit aversive and threatening signals. Although VPMpc neurons are known to transmit taste signals, they found their novel role in aversive behaviors.

It is quite meaningful and beneficial to find molecularly defined neurons in VPMpc because the study of this brain area is not very active partly due to lack of molecular markers. On the other hands, the reviewer feels that additional experiments are required to characterize CCK/Oprm1 neurons before publication.

Main point

It is unclear whether CCK/Oprm1 neurons in VPMpc respond to gustatory stimuli including aversive and appetitive taste modalities. Especially, if the authors use the present title ("Gustatory Thalamic Neurons" Mediate Aversive Behaviors), they should test this point by in vivo calcium imaging.

It is critical to clarify whether aversive and gustatory stimuli are processed by the same or different VPMpc neurons. Although they seem to plan to publish the paper containing this point in line 300-316, I think they should show some of the evidence in the current manuscript, but not future one.

Minor points

1. According to Fig. 2a, they used AAV-DIO-GCaMPm:YFP. Is it correct?
2. The position of Fig. 2d and g is opposite.
3. In Extended Fig. 1, mCh should change to mCherry.

Version 1:

Reviewer comments:

Reviewer #1

(Remarks to the Author)

The new data suggest that albeit the small size of the VPMpc, it contains function- and/or projection-specific clusters of neurons.

Thank the authors for the detailed answers, clarifications and the additional dataset. The revised manuscript has gone through substantial changes.

I have no further comments.

Reviewer #2

(Remarks to the Author)

The authors have made substantial revisions to the manuscript that improve the paper.

Reviewer #3

(Remarks to the Author)

The revised manuscript is significantly improved, highlighted by new experiments including 1) validating that the VPMpc sends collateral projections to the IC and LA, 2) CCK neurons co-express with vGlut, and 3) a battery of experiments showing CCK neuron responses to various behaviors. The authors provided thorough responses to my concerns. I have just two minor comments.

1) A comment from the first submission noted elevated calcium activity before stimulus onset. The response from the authors pointed out that this was due to the experimenter's approach. But the new Extended Data 4 shows increases before the onset of sucrose and water lick, but not quinine. It also appears before the airpuff in Figure 2E. The reasons for this rise should be addressed in the discussion, particularly in relation to specific behaviors.

2) Line 382—383: "Indeed, we have shown using electrophysiology techniques that most of the same VPMpc neurons respond to photoactivation of axon fibers from either Satb2 or Calca neurons in the PBN (data not shown)." This is contradictory since the data is not shown. I don't think this sentence is necessary so I recommend removing it.

Reviewer #4

(Remarks to the Author)

The authors answered the reviewer's questions appropriately. I support their publication.

Point-by-point response (NCOMMS-24-56339-T, F. Cao et al.)

REVIEWER COMMENTS

Reviewer #1 (Remarks to the Author):

In this study, Cao et al provide series of evidence that the gustatory thalamus (VPMpc) has the ability to mediate aversive behavior. Using single cell Ca-imaging they show that VPMpc cells expressing CCK respond to thermal, tactile and electrical painful stimuli as well as to learned auditory signals. They also show that these cells can bidirectional control various forms of aversive behaviors.

The study has several strengths though also some limitations. Overall, the authors provide a valuable contribution to the field by expanding the list of neuronal mechanism for aversive behavioral responses with the gustatory thalamus.

Response: Thank you very much for your positive assessment of the paper. We appreciate your suggestions and comments, which we have addressed as follows.

My general comments and questions.

1) It was stated that “Thus, the Cck/Oprm1 neurons represent most, if not all, of the VPMpc neurons;...” . However, can it be really ruled out that there is a CCK-negative neuronal population which belongs to the gustatory thalamus?

Going further on this line, earlier studies – also cited by the authors, like Yasui et al., 1987 – showed that VPMpc also projects to the central amygdala. Recently, rabies-mediated tracing also showed prominent VPMpc-CeA connection (Douglass et al., 2017). According to the figure 6 and the extended fig 7, CCK cells do not project to the CeA. These could suggest that there is another (non-CCK) VPMpc cell group which targets CeA, and so, the CCK expression territory is only a subfield of the gustatory thalamus. One solution could be to examine the CCK-content of the insular cortical-projecting VPMpc cells. A large or total overlap between retrogradely labeled cells and CCK could provide a firm argument for investigating VPMpc based on the CCK expression.

Of course, even if there is another VPMpc population, the main conclusion of this study is still valid (if the neuronal perturbations using CCK-Cre line and Cre-dependent viral constructs were limited to VPMpc; see next comment) but only for the CCK population. Then, the title and the text should specify this.

Response: Thank you for the insightful comments. Based on the RNAscope data, *Cck*-expressing neurons occupied 52.86% of DAPI cells, which is like the ratio of all the neurons vs DAPI cells. We have also performed additional experiments to strengthen our conclusions. We added a new RNAscope experiment to examine the colocalization of *Cck* and *Slc17a6* (Vglut2) in the VPMpc, which is thought to be glutamatergic exclusively. As shown in Extended Data Fig. 2a, b, *Cck* mRNA was colocalized with *Slc17a6* mRNA in nearly all (96.96%) VPMpc neurons.

Please see **Extended Data Fig. 2** and **lines 94-98**.

When we identify the axonal projections from VPMpc, we also observed a weak projection to CeA, consistent with the studies by Yasui et al., 1987. It was not mentioned in the original manuscript because of the weak signal (about 1/10 of the LA signal). However, our new “collateral experiment” (Fig. 7a-c) revealed a stronger projection to the CeA. We discussed this issue in the revised manuscript and cite the Yasui et al., 1987 and Douglass et al., 2017 papers.

Please see **Fig. 7a-c**, **lines 294-296** and **lines 404-408**.

2) This is a methodically elegant study combining various behavioral tests, calcium imaging and in vitro electrophysiology with viral tracing. In Figure 6, the authors showed the results of a 15 nl AAV injection. Indeed, selective targeting of a small region like VPMpc requires that precision. Furthermore, the AAV signal amplification with antibody staining was also performed which is highly welcomed. However, in the rest of the experiments, 300 nL of AAVs were injected in the thalamus. In the view of the extended expression of CCK in thalamic areas (even in the neighborhood of VPMpc, including PIL/SPFp or posterior PVT), it is likely that this large volume of AAV transduced not only the VPMpc but also other thalamic territories. It would be necessary to show the size and regional distribution of the AAV infections (with antibody amplification) for each experiment.

Response: Thank you for the comment. We have added the bilateral VPMpc viral expression of both AAV1-DIO-ChR2:YFP and AAV1-DIO-YFP from a typical experiment in Extended Data Fig. 12. We also added higher magnification images showing VPMpc in two different bregma levels (-2.06 mm and -2.18 mm), in which VPMpc was labelled using white dashed lines. These images provide examples showing the size and regional distribution of the viral infections in VPMpc with 300 nL injections.

Please see **Extended Data Fig. 12**.

3) Based on the findings it is clear that VPMpc can also serve as US (or US-associated) signal for LA-dependent fear learning. However, it is also stated that “However, directly activating the

CCKVPMpc neurons was unable to serve as an unconditioned stimulus in the fear conditioning paradigm (F.C., unpublished).” Is it not contradiction?

Response: VPMpc neurons respond to US (or US-associated) signals. For example, different shock intensities and shock-associated fear memories increase the calcium activity in VPMpc neurons. However, directly activating the CCK^{VPMpc} neurons was unable to serve as an unconditioned stimulus in the fear conditioning paradigm. We have added this data to Extended Data Fig. 9. As shown in this figure, photo stimulation (30 Hz, 10 ms, 15 mW) of VPMpc neurons cannot substitute for a foot shock in the fear-conditioning paradigm. One explanation is that activating CCK^{VPMpc} neurons can enhance aversive responses but is not sufficient to serve as an unconditional stimulus. We did this experiment because activation of CGRP^{PBN} axon terminals in the VPMpc does substitute for a US foot shock in a fear-conditioning paradigm (Bowen et al., 2020) This is likely due to activation of fibers of the central pathway that connects the PBN to the extended amygdala and passes just below the VPMpc. The results emphasize the importance of direct activation of VPMpc neurons.

Please see **Extended Data Fig. 9 and lines 237-240.**

It is interesting that VPMpc cells only projects to the rostral part of the LA. Does it mean that rest of the LA receives US-related signals from other sources? Indeed, certain MGN regions and neighboring thalamic areas (like PIL/SPFp) has been shown to send associative/multisensory aversive signal rather than pure US to the LA (Barsy et al 2020, Taylor et al, 2021, Kang et al., 2022) without any topography. Barsy et al (2020) showed that a calcium binding protein, the calretinin (CR) marks most of the thalamic territories including gustatory thalamus which send projection to the LA. Furthermore, CR thalamic cells also heavily project to layer 1 of insular and other associative cortical areas similar to VPMpc. Thus, VPMpc shares many similarities with PIL/SPFp.

Thus, a question could arise why distinct thalamic regions projects to LA. Do they code slightly distinct elements (features) of the environment and provide these as associative signals (feature1-US, feature2-US, ect) to the LA? Some sort of association can already take place before LA, at thalamic or even upstream to it, at the superior colliculus or parabrachial nucleus level.

Response: Thank you for the comments. These are great questions. It’s possible that the medial and posterior LA receive US-related signals from other brain regions, including the local circuits coming from amygdala itself. We cannot rule out the possibilities of other US inputs to rostral LA. Reciprocal local circuits within amygdala and lateral amygdala may also contribute to the US pathways. VPMpc not only carries multisensory aversive modalities, but also is the main thalamic nucleus relaying gustatory input to the gustatory cortex. We recorded the individual

neuronal responses to seven different stimuli (water, sucrose, quinine, air puff, clapping, tail lifting and tail pinch) and observed that some neurons that respond to taste stimuli also respond to aversive sensory modalities, also without any obvious topography. Therefore, we suspect the information carried from distinct thalamic regions to LA, as well as the VPMpc itself to its descending pathways could be encoding distinct elements, a combination of multiple features or a mixture of both, possibly depending on the upstream inputs. For example, VPMpc receives inputs from PBN. PBN also encodes multisensory aversive sensory and taste modalities (Campos et al., 2018, Jarvie et al., 2021, Kang et al., 2022). The role of rLA is not fully studied. Part of the reason is due to the technical difficulties in targeting the region without any molecular markers or clear borderlines. We have added new data showing that photoactivation of the VPMpc projection to the rLA region is able to promote long-lasting allodynia.

Please see **Fig. 7l, m; Extended Data Fig. 5; lines 149-157 and lines 314-318.**

Other comments and questions:

a) Most of avg Ca transients start to elevate before the onset (Fig 2). What is the explanation for that?

Response: Thank you for the comment. The different aversive sensory stimulations were manually performed by the experimenter in the behavior room. The increasing Ca^{2+} transients before the onset of stimulation is likely due to the transient disturbance from the experimenter approaching the cage and touching the mice. This is consistent with less disturbance with hand clapping since the experimenter does not need to touch the mice in that test.

b) Fig 2 and 3. The avg Z score values are between 1 and 2, except in the cases of foot shock trials. Still, the scaling of the heat maps is from -6 to 6 everywhere (or even larger in the extended figures), except in the US panels, where it is -4 and 4. Why is this different scaling? For valid comparisons, the same scaling would be preferred.

Response: Thank you for the comment. The same scaling is now provided in all the figures.

4) The n numbers for imaged cells (in Fig 2-3) are varying from one experiment to other although these were collected from the same animals. What is the reason for that?

Response: Thank you for the comment. Yes, the cell number for aversive sensory stimuli (Fig. 2c, f, i, l), foot shocks (Fig. 2p-r) and fear learning (Fig. 3g, h, i) are different. This is due to the difference in the number of mice that were used in conducting the Ca^{2+} recordings. Some of the experiments used the same group of mice while others used more mice. In total, we recorded 4 mice for aversive sensory stimuli, 2 mice for foot shocks and 3 mice for fear learning. Although

the total number of mice used in these three experiments were different, the number of cells in aversive sensory stimuli (Fig. 2c, f, i. l) and foot shocks (Fig. 2p-r) are the same.

5) Fig 2 and 3. In what order are the cells placed on the heat map. Were those always placed in the same order?

Response: Only the cells in the foot shock paradigm (Fig. 2p-r) were in the same order. The recording under three shock intensities were performed on the same day in the same trial. Each line in the heatmap represents the same cell across the three different shock intensities. The cells in other groups were placed randomly since some of the stimuli were performed on a different day. We have added new data comparing the same cell across different aversive stimuli (Extended Data Fig. 5). This new heatmap shows individual neuronal responses of each cell exposed to air puff, clapping, tail lifting and tail pinch.

Please see **Extended Data Fig. 5 and lines 149-151.**

6) As all of these experiments are performed using the same animals, is it possible to track the same cell throughout the all experiments? This type of analysis could reveal whether the same cell can code various forms of aversive signals or there are stimulus specific groups of CCK VPMpc cells.

Furthermore, tracking the same cell throughout the 3 days of fear-learning paradigm, could show the presence of any plastic changes of thalamic cells previously shown in other, amygdala projecting thalamic population (Taylor et al, 2021, Barys et al 2020).

Response: The mice number for these experiments were not the same. We did not perform the longitudinal tracking methods in that experiment because there were not enough cells that could be confidently tracked across three different days. In total, we recorded 4 mice for aversive sensory stimuli, 2 mice for foot shocks and 3 mice for fear learning. The cells tracked under the foot shock paradigm (Fig. 2p-r) were the same cells throughout the three different shock intensities. Each line represents the same cell. We have added Extended Data Fig. 5 to compare the same cell response to different sensory stimuli, which demonstrates the coding pattern of various forms of aversive signals. Although the fear-learning paradigm was performed using the same mice across three days, the cells in each day were placed randomly since the field of view changed when the miniscope was reinstalled.

Please see **Extended Data Fig. 5 and lines 149-157.**

7) Fig 5c. Either the y axis title or the scaling is not correct.

Response: Thank you for pointing this out. Fig 5c have been corrected.

8) Is there any data, either in the literature or in the authors' hand, showing the projection-selectivity of the CCK VPMpc cell? Whether the IC and LA projecting cells are the same or belong to distinct VPMpc subgroups? If the CCK VPMpc cells have convergent projection to IC and LA, then, the observed behavioral changes (with TetTox, hM3Dq, ChR2) can be due to either circuit. Otherwise, as it is mentioned in the discussion, projection-selective perturbation is required to reveal the existence of functional specificity of the VPMpc-LA and VPMpc-IC pathways.

Response: Thank you for this insightful comment. We agree with this reviewer that identifying the subgroups of IC and LA projecting neurons in VPMpc, as well as the potential function of the distinct VPMpc-IC and VPMpc-LA pathways is important. We have now added the following experiments:

1) We examined whether VPMpc-IC and VPMpc-LA are from a collateral pathway by injecting AAVretro-FLP:zsGreen to IC and AAV-Cre_{on}Flp_{on}:mCherry to VPMpc (Fig. 7b). This experiment revealed that VPMpc neurons send collaterals to both the IC and rLA (Fig. 7c).

2) We examined the circuit function of VPMpc-IC and VPMpc-LA by injecting ChR2 into VPMpc and placing a fiberoptic over the IC or LA (Fig. 7d, k). Photoactivation of VPMpc>IC or VPMpc>LA terminals induced long-lasting allodynia. Activation of VPMpc>IC was aversive (based on RTPP assay) while activation of VPMpc-LA pathway was not. These new experiments have also been added to the manuscript (Fig. 7d-q).

Please see **Fig. 7**.

Reviewer #2 (Remarks to the Author):

Does activation/inhibition of these neurons induce baseline changes in locomotion? The open field experiments suggest that activation may increase locomotion and inhibition may decrease locomotion, which could have significance in interpreting many of the other outcomes suggested to be a sign of aversion.

Response: Thank you for the comments. hM3Dq chemogenetic activation and TetTox inhibition of CCK/OPRM1^{VPMpc} neurons did not affect locomotion. Photoactivation using 20 Hz, 10 ms light stimulation increases the locomotion and induces aversion in mice. However, we don't have evidence to show that this increased locomotion is related to their aversive behaviors.

For recording experiments in VPMpc neurons: in almost all cases the effects on activity are pretty subtle and there are as many neurons that decrease their activity as increase, yet this is not mentioned or discussed.

Response: Thank you for the comment. The calculation of average trace and heatmaps are based on the Z score of the calcium fluorescence. The activation, inhibition or unresponsive cell identification were based on the area under the curve (AUC). Neurons were deemed as 'increased' if the AUC of post-stimulation has > 50% increase compared to the AUC of pre-stimulation. Neurons were deemed as 'decreased' if < 50% and deemed as 'no response' if the difference of AUC did not show > 50% increase or decrease. The average z scores for average sensory responses ranged from 1 to 1.5, like the average z score value induced by 0.3-mA foot shock. The average z score for 0.5-mA foot shock is close to 4. The average z score indicates the total value of all neurons, including the 'increased', 'decreased' and 'no response' neurons.

How this pattern of activity (some decrease and some increase) maps onto the gain and loss of function experiments could be discussed too.

Response: Thank you for the comment. The single cell calcium imaging was designed to demonstrate the dynamic activities of the individual neurons in response to multiple different stimulation modalities. Chemogenetic hM3Dq activation, optogenetic photo-activation and TetTox inhibition of CCK/OPRM1^{VPMpc} neurons were used to pursue the gain- and loss-of-function experiments. In those experiments we did not attempt to measure calcium dynamics in single cells or the entire VPMpc. We only measured calcium dynamics in response to transient stimuli.

Do salient but non-aversive experiences increase activity of these neurons?

Response: Thank you for asking this question. We have also tested the calcium activity of VPMpc neuron while mice were given 5% sucrose. This non-aversive experience also increased the activity of VPMpc neurons. We have also added this description into the manuscript.

Please see **Extended Data Fig. 4b, e** and **lines 144-147**.

Several of these interventions would induce activity in brain areas that would not be classified as mediating aversion (clapping induces calcium signals in V1 neurons for example). The fear learning result ameliorates this a little, but are the authors sure that the neural responses are due to aversion rather than surprise or startle?

Response: Thank you for this comment. We agree with this reviewer that many sensory stimuli used in the manuscript may also elicit a surprise or startle to mice. We have added one more experiment to show the aversive response generated by giving mice quinine, which also activates VPMpc neurons without generating a startle response.

Please see **Extended Data Fig. 4c, f** and **lines 144-147**.

As the stimuli provided are all likely to induce some kind of startle and/or motion are the authors confident that the increases/decreases are not due to a shift in the imaging z-plane.

Response: Thank you for the comment. We have employed rigid motion correction algorithm provided by IDPS to the calcium recording images to minimize motion artifacts during analysis. When we looked at the recording frame after motion correction, calcium signals were aligned with stimulation without shifting the z-plane. We also recorded the calcium activity change while mice were tested in a lickometer without obvious motion changes. Our results also showed the calcium activity changes while giving a certain taste stimuli mice were not moving.

Please see **Extended Data Fig. 4** and **lines 144-147**.

Figure 2 j - this plot is incorrect (values don't match size of shaded area)

Response: Thank you for pointing this out. We have fixed this figure.

Minor comments:

-In Figure 2 heatmaps. The colorbar axes are too broad to fully appreciate the activity changes in most neurons. I could understand if the authors were trying to fix the axes of all heatmaps but they do differ in c vs p for example.

Response: Thank you for this comment. We have now adjusted the color bar axes so that all the panels are consistent.

In Figure 2 b,e,h,k - is this the average of all recorded neurons? Text in figure legend is not clear.

Response: Thank you for this comment. The data in Figure 2 b,e,h,k represent the average of all recorded neurons under each aversive sensory stimulus. We have now added text information in the figure legend.

Please see **line 953**.

Reviewer #3 (Remarks to the Author):

This manuscript titled “Gustatory Thalamic Neurons Mediate Aversive Behaviors” examines the role of thalamic subnuclei circuitry in fear learning and threat behavior. The VPMpc is generally thought to be involved in taste aversion, but the authors argue it may play a more prominent role in aversive behaviors. The manuscript elegantly demonstrates that Cholecystokinin (Cck) and mu-opioid receptors (Oprm1) are co-expressed in the same VPMpc neurons, and these neurons can be manipulated to alter aversive behaviors. The manuscript is well-written, and the experiments are technically innovative. However, there are several weaknesses and concerns that detract from my enthusiasm. The hypothesis is that the VPMpc transmits aversive signals to the IC to mediate aversive behaviors, but this isn’t directly tested. The advantage and significance of using Cck and Oprm1 as molecular markers over examining the PBN-VPMpc-IC circuit independent of molecular markers is unclear. For example, is the goal to target Cck ultimately pharmacologically and/or Oprm1? This concern is compounded by the inconsistent use of Cck-Cre and Oprm1-Cre mice throughout the manuscript. In addition, I have larger concerns about the lack of statistical rigor, as presented. In all, the manuscript has the potential to have a high impact given the novelty of the VPMpc playing a role in non-taste-related aversive behavior. Still, this manuscript is somewhat disjointed at times and would benefit from a more focused narrative and better-outlined goals and conclusions. Specific comments are summarized below:

Response: Thank you for the positive assessment of the paper as well as the insightful comments. We have added new functional studies to specifically target VPMpc-IC circuit using an optogenetic strategy. The new data shows that photostimulation of VPMpc-IC terminals can induce chronic pain, aversive response and aversive memory. Cck and Oprm1 were identified as molecular markers of VPMpc neurons that receive direct inputs for parabrachial *Calca* neurons. Identifying the molecular marker is one of the essential steps to specifically study the PBN-VPMpc-IC circuit. We currently only use *Cck-Cre* and/or *Oprm1-Cre* mice and AAV carrying Cre-dependent effector genes to manipulate and trace VPMpc neurons. We show that Cck and Oprm1 are co-expressed in all cells. We tested each individually, both mouse lines showed the same histology and behavior results so that the data obtained from both were merged. We appreciate the detailed specific comments which we have addressed as follows.

1)The justification for interchangeably using both mouse lines is lacking. Some experiments use all of one strain, and some use a mix, which is a concern. It seems like focusing on one mouse line would be more efficient from a narrative perspective. Examples:

a.Line 85-88: “Fos was expressed in >18% of either Cck- or Oprm1-expressing neurons...” but

the figure legend for 1h says the data is just Cck neurons.

b. Figure 4: A-D is CCK-Cre, E-J is a mix, and K-M is Oprm1-Cre.

Response: Thank you for this comment. *Cck-Cre* and *Oprm1-Cre* mice were both used in this study because our first step of this study was focusing on identifying the molecular markers of VPMpc neurons that innervated with CGRP^{PBN} neurons. We have tested different Cre-driver lines of mice (*Calcr1^{Cre}*, *Ntsr1^{Cre}*, *Adcyap1r1^{Cre}* or *Tacr1^{Cre}*) for receptors of the neuropeptides made by CGRP^{PBN} neurons but none of them were the neuronal markers of VPMpc neurons. Luckily, we found *Cck* and *Oprm1*. The initial experiments were conducted using these two Cre-driver lines of mice in parallel since we did not know if they were representing the same or different subtypes of VPMpc neurons and may have distinct behavior functions. After we gathered pilot data, we found that these two neuronal markers were very similar and represented the same behavior phenotypes. We carefully compared results between these two Cre-driver lines of mice and performed the RNAscope to measure the overlap that we found CCK^{VPMpc} and OPRM1^{VPMpc} neurons are the same neurons. After this milestone, we focused on the *Cck^{Cre}* mice for the rest of the study. These are the reasons why some of the experiments were done in *Cck^{Cre}* mice and some of them were conducted in both strains. We wanted to present the original data from all the mice that were tested and therefore combined the data from both *Cck^{Cre}* and *Oprm1^{cre}* mice. For the colocalization of *Fos* and *Cck/Oprm1*, we only included the colocalization figure in Cck neurons (Fig. 1h). We did both calculations that *Fos* was expressed in 18.39% of Cck-expressing neurons and 18.44% of Oprm1-expressing neurons so that we describe in the text as ‘*Fos* was expressed in >18% of either Cck- or Oprm1-expressing neurons’. Because Cck and Oprm1 neurons are the same cells, we only included the *Fos* colocalization figure in Cck neurons.

2) The authors argue that the Cck/Oprm1 expressing neurons represent “most, if not all of the VPMpc neurons” (Line 269). I’m not grasping the significance of these molecular markers if essentially the entire VPMpc (at least the neurons) is represented.

Response: Thank you for this comment. We added a new experiment that verified nearly all VPMpc neurons are Cck/Oprm1 expressing neurons. This is also important because we can use Cck/Oprm1 as a neuronal marker to target the entire VPMpc neurons. VPMpc is a relatively small parvicellular part of the ventral posteromedial nucleus. Local cannula injection or lesion cannot specifically target this small region. However, with this molecular neuronal marker, we can specifically target VPMpc neurons, as well as using this molecular marker to explore the upstream and downstream circuit functions.

Please see **Extended Data Fig. 2** and **lines 94-98**.

3) If I understand the data correctly, fos was expressed in 18% of Cck+ neurons after CGRP chemogenetic activation. As this is a marker of increased neuronal activity, it fits nicely with the mini-scope data showing a similar percentage of cells that increase during the behavior. But is fos expressed in neurons that do not express Cck? What is the significance of the 82% of Cck neurons that do not express fos? This experiment could generally use more detailed analysis and discussion of results and interpretation.

Response: Thank you for the comment and these questions. All the Fos-expressing neurons are Cck-expressing neurons. Chemogenetic activation of CGRP^{PBN} neurons activated 18% of CCK^{VPMpc} neurons; hM3Dq and CNO never activates more than a fraction of cells in which it is expressed.

4) Figure 1E shows data points with over 100% colocalization. I'm not sure how this is possible.

Response: Thank you for pointing this out. This has been corrected.

5) Figure 2 shows bulk fluorescence increases, but the percentage of cells increasing is small. Are these the same cells? From the heatmaps, it looks like generally no.

a. Are the percentages similar between animals?

b. Was foot shock performed in separate mice?

c. Why does the number of cells differ between tests? The n is 76 for startle, 29 for FS, and then 59 for FC in figure 3.

Response: Thank you for these questions. The calculation of average trace and heatmaps are based on the Z score and the activation, inhibition or unresponsive cell identification were based on AUC. The mice number for these experiments was not completely the same but the percentages are similar between animals. Foot shocks were not performed in separate mice. Mice that received foot shocks also received other aversive sensory stimuli. The number of cells differs between tests because of the different animal numbers in each test. In total, we recorded 4 mice for aversive sensory stimuli, 2 mice for foot shocks and 3 mice for fear learning.

6) Lack of statistical comparison. Line 148- "the response was much greater than the response on day 1". The area under the curve during and/or after the tone should be measured to make this conclusion.

Response: Thank you for this comment. We have measured the area under the curve (AUC) during the tone and after the tone each day. We also performed the Mann Whitney test and compared the AUC during each phase. The AUC on Day 2 after the tone is significantly higher

than the one on Day 1 ($P < 0.001$). The AUC during the tone and after the tone on Day 3 does not have statistically difference but both showed an increase trend compared to Day 1.

7) In general, there needs to be more statistical rigor throughout. Many tests are just reported as two-way ANOVAs with very little detail. Effects should be reported for interactions, group, treatment, post-hoc comparisons, etc. For example, the CPP in Figure 4m is within-subject, so the data is represented twice (70% on the saline side is the same as 30% on the U50 side). The only comparison should be YFP vs. TetTox on either the saline or U50 sides.

Response: Thank you for this comment. We have provided the n number, analysis type, F value, as well as the post-hoc comparisons (Holm-Sidak's multiple comparisons) for two-way ANOVA tests. All the individual statistical analysis were individually listed in the legend. We now have replaced these and added more details and included all statistical analysis into a separate form as Extended Data Table. 1 at the end of this manuscript. For the Fig. 4m, we separated the CPP analysis to YFP and TetTox group and showed the percentile of preference in each chamber. Paired t test was used to compare the saline side and the U50 side in both groups. This figure is consistent with the panels in Fig. 4l, 5l-n, 7g-j, n-q so that we did not modify but the detailed statistical analyses were added for all of them.

Please see **Extended Data Table. 1.**

8)The data in Figures 2 and 3 suggest a shift from neurons that decrease activity to neurons that increase activity. This is an exciting finding, but Figure 4 broadly inhibits all Cck/Oprm neurons, not just the percentage of neurons that increase activity. How do these findings relate? More discussion comparing results across figures is needed.

Response: Thank you for the comment. Figure 4 shows a broad inhibition using TetTox and the results induced by TetTox are net effects of all the neurons. These neurons include individual neurons with dynamic activities in Figure 2 and 3. We agree that we cannot equalize the net effect of TetTox inhibition to the dynamic changing of the individual neurons. The increase or decrease of individual neuron activity is determined by different stimulation modalities and may change moment to moment and day to day. Our purpose is to show that the VPMpc neurons respond to aversive sensory stimulation and contribute to the fear-learning memory recall, as well as providing a general idea of what can activate/inactivate them.

9)The authors identify projections from the VPMpc to the insula, but this is already an established bidirectional pathway. A functional role for this pathway in aversive behaviors would strengthen the manuscript.

Response: Thank you for the comment. We have added new functional studies to specifically target VPMpc>IC circuit using optogenetic strategy. This new data shows that photostimulation of VPMpc>IC terminals induce chronic pain, aversive response and a short aversive memory.

Please see **Fig. 7d-j** and **lines 297-314**.

10) Similarly, electrophysiology establishes the functional connection between CGRP neurons and Cck/Oprm1 in the VPMpc. Still, the author does not present data demonstrating that activating CGRP terminals in the VPMpc affects aversive behavior.

Response: Thank you for the comment. A previous paper from the lab (Bowen et al., 2020) showed that photo-stimulating CGRP terminals in the VPMpc induced aversive response and promoted associative fear learning. Mice avoid photo-stimulation of CGRP^{PBN} neuron terminals in the VPMpc in the RTPP test.

11) Do all VPMpc neurons project to the LA and the IC (collaterals)? Or do subpopulations of neurons project to either the IC or LA? This distinction is essential, considering the tet experiment targets all Cck or Oprm1 neurons. Retrograde labeling from the IC and LA and assessing colocalization in the VPMpc would address this.

Response: Thank you for the comment. We have performed additional experiments showing that VPMpc>IC and VPMpc>LA are from a collateral pathway. We injected AAVretro-FLP:zsGreen into IC and AAV-Cre_{on}Flp_{on}-mCherry in the VPMpc of *Cck^{Cre}* mice (Fig. 7b). IC-projecting CCK^{VPMpc} neurons were labeled as red and we observed mCherry signals in both IC and rLA and adjacent areas. These results indicate that IC-projecting VPMpc neurons also project their terminals to rLA which implies collaterals to both (Fig. 7c).

Please see **Fig. 7 a-c** and **lines 284-294**.

12) Lines 287-288 show that directly activating CCK neurons was unable to serve as an unconditioned stimulus to fear conditioning. This seems like exciting data that should be included.

Response: Thank you for this comment. We have added these data to the manuscript.

Please see **Extended Data Fig. 9**.

13) Minor- Extended Figure 1 mentions Crhr1-Cre, but these data are not shown.

Response: Thank you for pointing this out. This oversight has been fixed.

Please see **line 1118**.

14) Minor- Lines 311-314 referencing data “to be published elsewhere” should be removed as the data cannot be verified.

Response: Thank you for the comment. This sentence has been removed in the manuscript.

Please see **line 383**.

Reviewer #4 (Remarks to the Author):

Cao et al., identified neurons expressing cholecystokinin (CCK) and mu opioid receptor (Oprm1) in the parvocellular part of the ventral postmedial nucleus (VPMpc) of the thalamus transmit aversive and threatening signals. Although VPMpc neurons are known to transmit taste signals, they found their novel role in aversive behaviors.

It is quite meaningful and beneficial to find molecularly defined neurons in VPMpc because the study of this brain area is not very active partly due to lack of molecular markers. On the other hands, the reviewer feels that additional experiments are required to characterize CCK/Oprm1 neurons before publication.

Response: Thank you very much for your positive assessment of the paper. We appreciate your suggestions and comments and added the requested additional experiments. Please see them below.

Main point

It is unclear whether CCK/Oprm1 neurons in VPMpc respond to gustatory stimuli including aversive and appetitive taste modalities. Especially, if the authors use the present title (“Gustatory Thalamic Neurons” Mediate Aversive Behaviors), they should test this point by *in vivo* calcium imaging.

Response: Thank you for the comment. We have added new *in vivo* calcium imaging experiments showing CCK^{VPMpc} neurons also respond to gustatory stimuli including water, sucrose and quinine. Sucrose was used as appetitive and quinine was used as aversive taste modality. We recorded CCK^{VPMpc} neuron activity during voluntary licking for these three different taste modalities and observed increased calcium activity in response to all these gustatory stimuli. We have included this data in the manuscript.

Please see **Extended Data Fig. 4** and **lines 144-147**.

It is critical to clarify whether aversive and gustatory stimuli are processed by the same or different VPMpc neurons. Although they seem to plan to publish the paper containing this point in line 300-316, I think they should show some of the evidence in the current manuscript, but not future one.

Response: Thank you for this comment. We have added data showing that aversive sensory stimuli and gustatory stimuli can be processed by the same VPMpc neurons. We alternately applied three taste stimuli (water, sucrose and quinine) and four aversive sensory stimuli (air puff, clapping, tail lifting and tail pinch) to mice and recorded 39 individual neuronal responses in response to these stimuli modalities (Extended Data Fig. 5a). Many neurons respond to more than one stimulus modality and some neurons respond to taste stimuli also respond to aversive sensory modalities. There is no clear evidence showing that different clusters of neurons are preferentially responded to a certain aversive or gustatory stimulus modality (Extended Data Fig. 5b, c).

Please see **Extended Data Fig. 5** and **lines 147-157**.

Minor points

1. According to Fig. 2a, they used AAV-DIO-GCaMPm:YFP. Is it correct?

Response: Yes, the virus used in calcium imaging is AAV-DIO-GCaMP6m:YFP.

2. The position of Fig. 2d and g is opposite.

Response: Thank you for pointing this out. This oversight has been fixed.

3. In Extended Fig. 1, mCh should change to mCherry.

Response: Thank you for this comment. This has been fixed.

Point-by-point response (NCOMMS-24-56339-A, F. Cao et al.)

REVIEWERS' COMMENTS

Reviewer #1 (Remarks to the Author):

The new data suggest that albeit the small size of the VPMpc, it contains function- and/or projection-specific clusters of neurons.

Thank the authors for the detailed answers, clarifications and the additional dataset. The revised manuscript has gone through substantial changes.

I have no further comments.

Response: Thank you very much for your positive assessment of the revised manuscript.

Reviewer #2 (Remarks to the Author):

The authors have made substantial revisions to the manuscript that improve the paper.

Response: Thank you very much for your positive assessment of the revised manuscript.

Reviewer #3 (Remarks to the Author):

The revised manuscript is significantly improved, highlighted by new experiments including 1) validating that the VPMpc sends collateral projections to the IC and LA, 2) CCK neurons co-express with vGlut, and 3) a battery of experiments showing CCK neuron responses to various behaviors. The authors provided thorough responses to my concerns. I have just two minor comments.

1) A comment from the first submission noted elevated calcium activity before stimulus onset. The response from the authors pointed out that this was due to the experimenter's approach. But the new Extended Data 4 shows increases before the onset of sucrose and water lick, but not quinine. It also appears before the airpuff in Figure 2E. The reasons for this rise should be addressed in the discussion, particularly in relation to specific behaviors.

Response: Thank you for the comment. We still believe the early onsets were due to the experimenter's approach. If we compare the onsets in Fig. 2b, e, h, k, clapping is less interventive than air puff, pinch and tail lifting, which is likely due to the indirect contact

between the experimenter and mice during the experimental onsets. The more precise onsets in Fig. 3 also support this thought since tone and shock were controlled by the computer without any disturbance from the experimenter. For the taste solutions in new Extended Data 4, mice tend to prefer quinine less due to its bitter attribute. Unlike the immediate sipping of water and sucrose upon the sipper tube insertion, mice took longer to start licking quinine, which results in less interference from insertion of the that taste tube into the cage. The amount of the 'onset earliness' in Extended Data 4 also aligned with the overall quickness of sipping tube licking: sucrose > water > quinine. Therefore, we believe the inconsistency of the early onsets were due to the experimenter's approach.

2)Line 382—383: "Indeed, we have shown using electrophysiology techniques that most of the same VPMpc neurons respond to photoactivation of axon fibers from either Satb2 or Calca neurons in the PBN (data not shown)." This is contradictory since the data is not shown. I don't think this sentence is necessary so I recommend removing it.

Response: Thank you for the comment. We have removed this sentence in the manuscript.

Reviewer #4 (Remarks to the Author):

The authors answered the reviewer's questions appropriately. I support their publication.

Response: Thank you very much for your positive assessment of the revised manuscript.